# Turbulence effects on warm rain formation in precipitating shallow convection revisited

Axel Seifert[1] and Ryo Onishi[2]

[1]Deutscher Wetterdienst, Offenbach, Germany
[2]Center for Earth Information Science and Technology, Japan Agency for Marine-Earth Science and Technology, Yokohama Kanagawa, Japan

*Correspondence to:* Dr. Axel Seifert (axel.seifert@dwd.de)

**Abstract.** Two different collection kernels which include turbulence effects on the collision rate of liquid droplets are used as a basis to develop a parameterization of the warm rain processes autoconversion, accretion and selfcollection. The new parameterization is tested and validated with help of a 1D bin microphysics model. Large-eddy simulations of the rain formation in shallow cumulus clouds confirm previous results that turbulence effects can significantly enhance the development of rain water in clouds and the occurrence and amount of surface precipitation. The detailed behavior differs significantly for the two turbulence models revealing a considerable uncertainty in our understanding of such effects. In addition, the large-eddy simulations show a pronounced sensitivity to grid resolution which suggests that besides the effect of sub-grid small scale isotropic turbulence which is parameterized as part of the collection kernel also the larger turbulent eddies play an important role for the formation of rain in shallow clouds.

## 1 Introduction

The formation of rain in warm liquid clouds is a result of the condensational growth on cloud condensation nuclei, and the subsequent growth of these droplets by binary collisions (Beard and Ochs, 1993; Pruppacher and Klett, 1997; Beheng, 2010). Especially in strongly turbulent clouds, like cumulus convection, the in-cloud turbulence can potentially increase the frequency of such binary collisions and thereby enhance rain formation (Devenish et al., 2012; Grabowski and Wang, 2013). This problem has attracted considerable attention over the last two decades culminating in the formulation of the semi-empirical collision-coalescence kernel of Ayala and Wang (Ayala et al., 2008b, a; Wang et al., 2008). This collection kernel attempts to provide a complete and quantitative description of the collision processes in turbulent (warm) clouds. Subsequently, Seifert et al. (2010) have applied this kernel and formulated a two-moment bulk microphysical model that takes into account the turbulence effects on autoconversion and accretion as predicted by the Ayala-Wang kernel. In large-eddy simulations (LES) of trade wind cumulus convection Seifert et al. (2010) have shown a significant impact of the turbulence effect on in-cloud rain formation and surface rain rates. These results, which were based on a two-moment bulk scheme, have later been largely confirmed by Wyszogrodzki et al. (2013) using a bin microphysics model in an LES.

The semi-empirical collision-coalescence kernel of Ayala and Wang is to a large extent based on the results of direct numerical simulation (DNS) which are necessary to quantify the turbulence effects on the collision statistics in terms of, e.g.,

the radial distribution function to describe the preferential concentration effect. As the DNS results are obtained at fairly low Reynolds number, much lower than observed within clouds, the formulation of the collection kernel includes an extrapolation to large Reynolds numbers. An alternative collection kernel recently proposed by Onishi et al. (2015) yields similar results at low Reynolds numbers where DNS data is available, but differs significantly in the Reynolds number dependency and the predicted values at high Reynolds numbers (Onishi and Seifert, 2016).

In the following we revisit the results of Seifert et al. (2010) and repeat most of their study, but now we apply the Onishi kernel and an updated version of the Ayala-Wang kernel. First, we derive and validate the corresponding two-moment bulk schemes, which already allows us some insights into the differences between the two kernels. Next, we apply the two bulk scheme in a large-eddy simulation study to test whether the differences between the two kernels matter in LES of trade wind cumulus clouds.

The structure of this paper very much follows in the steps of the Seifert et al. (2010) study. After a short review of the basic relations the two collection kernels are presented in section 2. In section 3 we use a box model to derive the enhancement factor for autoconversion. In section 4 the two-moment scheme is applied and validated in a 1D kinematic model. The large-eddy simulations are presented and discussed in section 5 followed by the Conclusions.

## 2 Parameterizations of the turbulence effects in the collision-coalescence kernel

For pure gravitational collisions the collection kernel can be written as (see e.g. Pruppacher and Klett, 1997)

$$K_{\text{grav}}(r_1, r_2) = \pi [r_1 + r_2]^2 \, |v(r_1) - v(r_2)| \, E_{\text{coll}} \tag{1}$$

where $r_1$ and $r_2$ are the radii of the two droplets, $v(r)$ is the terminal fall velocity of droplets, and $E_{\text{coll}}$ is the collision efficiency. For a turbulent flow the more general definition of the collision-coalescence kernel

$$K(r_1, r_2) = 2\pi [r_1 + r_2]^2 \, w_r \, g_{12} \, E_{\text{coll}} \, \eta_E. \tag{2}$$

has to be used. Here $w_r$ is the radial relative velocity at contact (Saffman and Turner, 1956, 1988). The radial distribution function $g_{12}$ quantifies the effect of preferential concentration on the pair number density statistics and $\eta_E$ represents an enhancement factor due to a modification of the collision efficiency by the turbulent flow. For further details and explanations of the basic concepts we refer to the recent reviews by Devenish et al. (2012) and Grabowski and Wang (2013).

Any physical model of $w_r$, $g_{12}$ and $\eta_E$ should be formulated in the dimensionless numbers that characterize the system. These are first of all the two Stokes numbers of the two colliding particles with the Stokes number being defined by

$$St = \frac{\tau_p}{\tau_k} \tag{3}$$

where $\tau_p$ is the particle relaxation time scale and $\tau_k$ is the Kolmogorov time scale. The particle relaxation time scale is given by

$$\tau_p = \frac{2}{9} \frac{\rho_p}{\rho_a} \frac{r^2}{\nu_a} \tag{4}$$

with the material density of the particle $\rho_p$ (here liquid water with $\rho_p = 10^3$ kg m$^{-3}$), the air density $\rho_a$ and the kinematic viscosity of air $\nu_a$. The Kolmogorov time scale $\tau_k$ is related to the Kolmogorov length scale $\ell_k$ and the turbulent dissipation rate $\epsilon$ by

$$\tau_k = \frac{\ell_k^2}{\nu_a} = \sqrt{\frac{\nu_a}{\epsilon}} \tag{5}$$

Due to the $r^2$-dependency of $\tau_p$ the Stokes number increases with droplet size. Typical cloud droplets with radii smaller than 20 $\mu$m have Stokes number below 0.2, large cloud droplets and small rain drops are close to $St = 1$, while larger raindrops have large Stokes number $St \gg 1$. Preferential concentration effects, i.e., large values of $g_{12}$, occur for $St \approx 1$. Smaller droplets with smaller Stokes numbers simply follow the flow and show no clustering, while drops with $St \gg 1$ do not feel the small scale turbulence due to their inertia and their trajectories are, in addition, largely determined by their significant terminal fall velocity. Therefore large cloud droplets and small raindrops with radii between 20 and 100 $\mu$m are most strongly affected by turbulence effects.

A turbulent flow is not yet fully characterized by $\tau_k$ (or $\epsilon$) alone. To quantify the root mean square of the turbulent velocity fluctuations, $u_{rms}$, we introduce the Taylor-microscale Reynolds number defined by

$$Re_\lambda = \frac{u_{rms}\lambda_T}{\nu_a} = \sqrt{15\frac{\nu_a}{\epsilon}}\frac{u_{rms}^2}{\nu_a}. \tag{6}$$

The Taylor-microscale Reynolds number is important for the collision statistics as it is closely related to the two-point correlation and the autocorrelation functions of turbulent flows. In general, turbulence has three independent length scales, the Kolmogorov scale, $\ell_k$, the Taylor microscale, $\lambda_T$, and a large-eddy or integral length scale (Pope, 2000). Therefore we will throughout most of this paper treat $\epsilon$ and $Re_\lambda$ as two independent variables. Only later when we apply the collision-coalescence model in LES we will parameterize $Re_\lambda$ as a function of $\epsilon$.

Various models have been suggested to parameterize $w_r$, $g_{12}$ and $\eta_E$ in terms of $St$ and $Re_\lambda$. Here we focus on the models of Wang and Ayala (Ayala et al., 2008b, a; Wang et al., 2008; Wang and Grabowski, 2009) and Onishi (Onishi, 2005; Onishi et al., 2015). A detailed discussion of these two models has recently been given by Onishi and Seifert (2016). We refer to those papers for the relevant parameterization equations. Figure 1 shows the enhancement factor of the collision kernel due to turbulence effects, i.e., the ratio $K(r_1, r_2; \epsilon, Re_\lambda)/K_{\mathrm{grav}}(r_1, r_2)$, for the Ayala-Wang and the Onishi model at $\epsilon = 1000$ cm$^2$s$^{-3}$ for two different values of $Re_\lambda$.

The Ayala-Wang model shows a significant increase of the collection kernel for high Reynolds numbers for droplet smaller than 80 $\mu$m radius, roughly a factor of 2 increase from $Re_\lambda = 1000$ to $Re_\lambda = 20000$ (Figs. 1a,b,c). For the Onishi kernel the $Re_\lambda$-dependency is more subtle and can be characterized as a shift of the maximum of the enhancement from smaller to larger droplets, i.e., the kernel decreases for small droplets ($r < 40$ $\mu$m) but increases for larger droplets ($r > 40$ $\mu$m) as the Reynolds number increases. For an in-depth discussion of the Reynolds number dependencies we refer again to Onishi and Seifert (2016).

# 3 Parameterization of turbulence effects on autoconversion

The evolution of the drop size distribution $f(x)$ as a function of drop mass $x$, where $f(x)dx$ is the number of drops per unit volume in the size range $[x, x+dx]$, is governed by the kinetic equation also known as the Smoluchowski coagulation equation (von Smoluchowski, 1916, 1917) which in its continuous form

$$\left.\frac{\partial f(x)}{\partial t}\right|_{koag} = \frac{1}{2}\int_0^x f(x-x')\,f(x')\,K(x-x',x')\,dx' - \int_0^\infty f(x)\,f(x')\,K(x,x')\,dx' \tag{7}$$

was first derived by Müller (1928). A detailed discussion of this equation and its mathematical properties is given in the classic review by Drake (1972) and more recently by da Costa (2015). Another classic, but still interesting contribution on the interpretation of the continuous form of the Smoluchowski equation is the paper by Gillespie (1975). Although various numerical methods are available to solve Eq. (7) directly (e.g. Berry and Reinhardt, 1974; Bott, 1998; Tzivion et al., 1999; Shima et al., 2009), this is most often seen as computationally too expensive in three-dimensional atmospheric models. Therefore bulk parameterizations are used which predict only a limited number of (partial) moments of the drop size distribution. Following Kessler (1969) and motivated by the emergence of bi-modal mass distributions as a consequence of the colloidal instability the size distribution is decomposed into two parts. Drops smaller than some threshold $x^*$ are called cloud droplets, drops larger than $x^*$ are called rain drops. The value of $x^* = 2.6 \times 10^{-10}$ kg which corresponds to a radius of 40 $\mu$m is not arbitrary but should be chosen as the local minimum of the bi-modal mass distribution function $g(x) = xf(x)$ during the colloidal instability (Beheng and Doms, 1986; Beheng, 2010). This minimum exists due to the properties of the (gravitational) coagulation kernel $K(x,y)$ which becomes less steep for $x > x^*$ (Long, 1974). Having defined the two drop categories, we can identify the following bulk microphysical processes: autoconversion is the formation of rain drops due to collisions between cloud droplets and accretion is the growth of rain drops due to the collection of cloud droplet by rain drops. The change of the number density within one category due to coagulation within this drop category is called selfcollection. For a more detailed review of the basic ideas of warm rain parameterizations we refer to the review of Beheng (2010). The increase in rain water content $L_r$ due to autoconversion and accretion is given by the integrals (Doms and Beheng, 1986; Beheng and Doms, 1986; Beheng, 2010)

$$\left.\frac{\partial L_r}{\partial t}\right|_{au} = \frac{1}{2}\int_{x'=0}^{x^*}\int_{x''=x^*-x'}^{x^*} f(x')\,f(x'')\,K(x',x'')\,x'\,dx''dx' \tag{8}$$

$$\left.\frac{\partial L_r}{\partial t}\right|_{ac} = \int_{x'=x^*}^{\infty}\int_{x''=0}^{x^*} f(x')\,f(x'')\,K(x',x'')\,x'\,dx''dx'. \tag{9}$$

For the parameterization of autoconversion we follow Seifert and Beheng (2001, SB2001 hereafter). For a cloud droplet distribution which initially obeys a gamma distribution in particle mass $x$

$$f(x) = Ax^\nu e^{-Bx} \tag{10}$$

SB2001 derived the autoconversion parameterization

$$\left.\frac{\partial L_r}{\partial t}\right|_{\text{au}} = \frac{k_{cc}}{20\,x^*}\frac{(\nu+2)(\nu+4)}{(\nu+1)^2}L_c^2\bar{x}_c^2\left[1+\frac{\Phi_{\text{au}}(\tau)}{(1-\tau)^2}\right]. \tag{11}$$

Here $L_c$ is the cloud water content, $\bar{x}_c = L_c/N_c$ the mean cloud droplet mass with $N_c$ being the cloud droplet number density, and $x^*$ is again the separating mass between cloud and rain drops. The dimensionless ratio $\tau = L_r/(L_c+L_r)$ with the rain water content $L_r$ acts as an internal timescale and modulates the autoconversion rate due to the universal function $\Phi_{\text{au}}(\tau)$ given by

$$\Phi_{\text{au}}(\tau) = 600\,\tau^{0.68}\,(1-\tau^{0.68})^3. \tag{12}$$

In case of purely gravitational collection the kernel parameter for autoconversion is given by $k_{cc} = k_{cc,0} = 9.44\times10^9 \text{ s}^{-1}\text{kg}^{-2}$ and originates from a piecewise polynomial approximation of the collection kernel (Long, 1974).

Following Seifert et al. (2010) we extend this autoconversion parameterization to include turbulence effects by making $k_{cc}$ a function of $\epsilon$, $Re_\lambda$ and $\bar{r}_c$. The latter dependency is necessary, because the turbulence effects are different for droplets of different size. Seifert et al. (2010) have shown that the Ayala-Wang kernel can be approximated with the following ansatz

$$k_{cc}(\bar{r}_c,\nu,\epsilon,Re_\lambda) = k_{cc,0}\left\{1+\epsilon\,Re_\lambda^p\left[\alpha_{cc}(\nu)\exp\left\{-\left[\frac{\bar{r}_c-r_{cc}(\nu)}{\sigma_{cc}(\nu)}\right]^2\right\}+\beta_{cc}\right]\right\} \tag{13}$$

where

$$\alpha_{cc}(\nu) = \frac{a_1+a_2\,\nu}{1+a_3\,\nu} \tag{14}$$

$$r_{cc}(\nu) = \frac{b_1+b_2\,\nu}{1+b_3\,\nu} \tag{15}$$

$$\sigma_{cc}(\nu) = \frac{c_1+c_2\,\nu}{1+c_3\,\nu} \tag{16}$$

are functions of the shape parameter $\nu$ only. Here we use the same ansatz for the updated Ayala-Wang kernel and for the Onishi kernel. The 11 coefficients of this model have been determined by a nonlinear least square fit using a data base of numerical solutions of the stochastic collection equation (SCE). The parameter space covered by the SCE simulations is $\epsilon \in [0,1000]$ cm$^2$ s$^{-3}$, $Re_\lambda \in [1000,25000]$, $L_c \in [0.2,2]$ g m$^{-3}$, $\bar{r}_c \in [8,20]$ $\mu$m and $\nu \in [0,4]$. Note that in contrast to Seifert et al. (2010) we have extended the range for $\epsilon$ to values up to 1000 cm$^2$s$^{-3}$ to allow for the higher dissipation rates that occur, for example, in cumulus congestus. The resulting coefficients for both turbulence kernels are given in Table 1.

The most notable difference between the two kernels is that for the Ayala-Wang kernel the autoconversion rate increases with $Re_\lambda$ resulting in $p = 1/4$, whereas autoconversion decreases slowly with increasing $Re_\lambda$ for the Onishi kernel with a power law exponent $p = -1/8$.

The different autoconversion enhancement factors for the two kernels and the quality of the fits is shown by Fig. 2 in which the Reynolds number dependency is also shown in more detail. The results for the Ayala-Wang kernel show somewhat higher enhancement factors compared to Seifert et al. (2010), mostly due to the improved treatment of the collision efficiency (cf. Onishi and Seifert, 2016). The Onishi kernel shows much lower enhancement factors and the maximum is shifted to larger (mean) droplet radii compared to the Ayala-Wang kernel. The $Re_\lambda$-dependency reveals that especially for the Onishi kernel the value of the exponent, $p = -1/8$, is not actually constant but the slope has significant dependencies on $\bar{r}_c$ and $Re_\lambda$. This more complicated behavior is consistent with the analysis presented by Onishi and Seifert (2016) who showed that the Reynolds number dependency of the kernel varies with Stokes number (e.g. their Figure 2). For the Ayala-Wang kernel the numerical data shows a slightly steeper increase with $Re_\lambda$ compared to the parameterization. This is mostly because we kept to exponent at $p = 1/4$ as in Seifert et al. (2010), although the extended range of the dissipation rate in the current study would call for a slightly higher exponent. The dependency on dissipation rate is assumed to be linear in Eq. (13) and this is confirmed for the Onishi kernel, but for the Ayala-Wang kernel the $\epsilon$-dependency becomes slightly weaker for high dissipation rates.

A first test of the autoconversion parameterization is obtained by simulations of exactly the same kind as used as training data, i.e., SCE simulations with an initial condition following a gamma distribution. As a metric for evaluation we use the time scale $t_{10}$ which is defined as the time needed to convert 10 % of the initial liquid water to rain water. Figure 3 shows the dependencies of $t_{10}$ on dissipation rate $\epsilon$, initial mean drop radius $\bar{r}_c$ and initial cloud water content $L_c$. This confirms that the fit is reasonable and that the autoconversion parameterization captures those dependencies correctly.

## 4 Turbulence effects in a 1D kinematic model

As in Seifert et al. (2010) we use the 1D kinematic model of Seifert and Stevens (2010) as a slightly more complete test problem for the warm rain scheme. The 1D kinematic model is especially useful as it describes the various stages of the warm rain formation in an isolated cumulus cloud. This is necessary to test and validate our assumptions regarding accretion and selfcollection of raindrops. Those two processes depend strongly on drop sedimentation and the resulting drop size distribution and can therefore hardly be tested in pure SCE simulations. Although the 1D model provides a reasonable idealized framework for such a test, we would recommend using a kinematic 2D model (e.g. Szumowski et al., 1998; Morrison and Grabowski, 2007) in future studies, because the 1d framework might not be sensitive enough to differences in the treatment of sedimentation which are more relevant in a more complex flow field. Here we apply the simpler 1D model for consistency with Seifert et al. (2010).

The accretion rate and selfcollection of rain are parameterized as

$$\frac{\partial L_r}{\partial t}\bigg|_{ac} = k_{cr} L_c L_r \Phi_{ac}(\tau)\, \eta_{ac} \quad \text{with} \quad \Phi_{ac} = \left(\frac{\tau}{\tau + 5 \times 10^{-4}}\right)^4 \tag{17}$$

and

$$\frac{\partial N_r}{\partial t}\bigg|_{sc} = -k_{rr} N_r L_r \eta_{sc} \tag{18}$$

with $k_{cr} = 5.78$ m$^3$ kg$^{-1}$ s$^{-1}$ and $k_{rr} = 4.33$ m$^3$ kg$^{-1}$ s$^{-1}$ and turbulent enhancement factors $\eta_{ac}$ and $\eta_{sc}$. In case of the Ayala-Wang kernel we use the same enhancement factors as in Seifert et al. (2010) with

$$\eta_{ac} = \eta_{sc} = 1 + \hat{c}_r\,\epsilon^{\frac{1}{4}} \tag{19}$$

with $\hat{c}_r = 0.05$ cm$^{-1/2}$ s$^{3/4}$. For the Onishi kernel we apply a stronger enhancement which is linear in the dissipation rate $\epsilon$

$$5 \quad \eta_{ac} = \eta_{sc} = 1 + \check{c}_r\,\epsilon\left(\frac{x^*}{\bar{x}_r}\right)^{\frac{2}{3}} \tag{20}$$

with $\check{c}_r = 0.8 \times 10^{-3}$ cm$^{-2}$s$^3$. For a dissipation rate of 1000 cm$^2$s$^{-3}$ this corresponds to an increase in accretion of 28 % in case of the Ayala-Wang kernel and 80 % for the Onishi kernel. For the Onishi kernel we have included an additional dependency on $\bar{x}_r = L_r/N_r$ to suppress the turbulent enhancement for very large (mean) raindrop sizes that do not feel the effect of small-scale turbulence. The enhancement factors for accretion and selfcollection cannot be directly derived from the collection kernel

alone. The turbulent enhancement of the collision rate leads also to changes in the drop size distribution, i.e., the increase in accretion and selfcollection is attributed, first, to the direct increase in the collision rates by the local turbulence and, second, to a modification of the drop size distribution by the turbulence effect. The latter constitutes a memory effect and makes it also difficult to discuss the turbulence effects on accretion and selfcollection separately, because these two processes are strongly linked. In the following we always mean the combined action of selfcollection of rain and accretion, when we discuss effects

of turbulence on the droplet growth by accretion.

Extensive test with the 1D kinematic model have shown that the parameterization compares reasonably well with the bin microphysics solution for both collection kernels. The most important metric to evaluate the warm rain scheme in the 1D kinematic model is the precipitation amount at the surface. One could argue that the timing is almost as relevant as the precipitation amount, but as shown by Seifert and Stevens (2010) the precipitation efficiency in the 1D cloud model depends mostly on the

time scales of dynamics and microphysics, respectively their ratio, the Damköhler number. Therefore we discuss here only the precipitation amounts which are presented in Figure 4 as a function of dissipation rate (which is assumed as homogeneous within the cloud) for two different Reynolds numbers and various aerosol number concentrations $N_a$. For further details, e.g., on the treatment of activation we refer to Seifert and Stevens (2010). For the Ayala-Wang kernel we find a significant increase in surface precipitation, for example, we find an increase by a factor of 2 for low $N_a$ (clean conditions) when $\epsilon$ is as large

as 1000 cm$^2$s$^{-3}$ compared to pure gravitational kernel ($\epsilon = 0$). For high $N_a$ the cloud does not produce any rain without the effect of turbulence on the collision rate ($\epsilon = 0$), but yields significant rain when turbulence can contribute to rain formation. For the Onishi kernel we find qualitatively the same behavior, but the rain amounts are significantly lower especially for low dissipation rates $\epsilon$. The different Reynolds number dependencies of both kernels are also visible in these surface rain amounts. For the Ayala-Wang kernel the rain amounts increase significantly for higher Reynolds numbers. In case of the Onishi kernel

a slight decrease is observed for high $N_a$ when increasing $Re_\lambda$ from 1000 to 20000. For $N_a = 50$ cm$^{-3}$ a slight increase with $Re_\lambda$ is visible for the spectral model, but not for the two-moment scheme. This can be attributed to the increase in the accretion rate in the Onishi kernel for high $Re_\lambda$ and this effect we have neglected in the bulk scheme (mostly because the Re-dependency is quite weak and in addition the low $Re_\lambda$ case is not important for cloud physics applications). Nevertheless,

the 1D kinematic model suggests that the turbulence effect on accretion is significant, and even more so in case of the Onishi kernel. Especially for low $N_a$, when autoconversion is quite efficient, accretion can become the limiting process for droplet growth and an increase in accretion due to turbulence effects can significantly affect surface rain amount. This will be further investigated using large-eddy simulations in the following section.

## 5  Turbulence effects in large-eddy simulations of trade wind cumuli

### 5.1  Model setup

To investigate the effect of in-cloud turbulence on rain formation in trade wind cumulus clouds we perform large-eddy sim-ulations of the Rain In Cumulus over the Ocean (RICO) case as described by van Zanten et al. (2011). We use the standard RICO case and not the moister initial condition as in Seifert et al. (2010). We apply the UCLA-LES model (Stevens et al., 2005; Stevens, 2007) on a domain of 51.2 km × 51.2 km with doubly-periodic boundary conditions, a simulation time of at least 30 h and a horizontal mesh size of 50 m with additional simulations at finer and coarser grid spacing. The model time step is variable with a maximum Courant number below 0.5. The time step is mostly dominated by the vertical grid spacing and velocity and approximately 1 s. The cloud microphysical parameterization follows SB2001 and Stevens and Seifert (2008) with the modifications described in the previous sections. For the shape parameter of the cloud droplet size distribution we use $\nu = 1$ in all simulations. The sub-grid scale (SGS) turbulence model is a Smagorinsky-Lilly closure including a proper treatment of anisotropic grids (Scotti et al., 1993). As described in detail in Seifert et al. (2010) the SGS models provides the local (grid point) turbulent dissipation rate $\epsilon$ which is needed for the turbulence effect on cloud microphysics. Additional assumptions are necessary for the Reynolds number $Re_\lambda$ as the SGS model does neither provide $Re_\lambda$ nor $u_{rms}$. Here we follow Wyszogrodzki et al. (2013) and parameterize $Re_\lambda$ as a function of $\epsilon$ alone. Consistent with homogeneous isotropic turbulence we use the scaling relation $Re_\lambda = Re_0(\epsilon/\epsilon_0)^{1/6}$ with $Re_0 = 10000$ and $\epsilon_0 = 100 \; \mathrm{cm^2 \; s^{-3}}$.

### 5.2  Turbulence effect on rain formation

Figure 5 shows time series from a first set of simulations with grid spacing $\Delta x = 50$ m. After some initial spin up the cloud liquid water path increases slowly with time corresponding to a slowly deepening cloud layer. Rain water develops after a few hours and surface precipitation is observed subsequently. The rain water path, surface rain rate and the timing of the rain formation differs strongly between the various simulations. The control simulation which uses the purely gravitational kernel develops only marginal rain and surface precipitation within the 30 h period. In contrast, the simulation which applies the Ayala-Wang kernel develops rain much earlier and the rain rate reaches 1 mm/d after about 20 h with some fluctuations later-on. Using the Onishi kernel leads to faster rain formation compared to the control simulations, but slower than for the Ayala-Wang kernel. At the end of the simulation period the Onishi kernel yields similar rain rates as the Ayala-Wang kernel, i.e., in the last hours both turbulence kernels increase the surface rain rate by a factor 7 relative to the control run. Especially for the Onishi kernel the enhancement of the rain formation is due to the combined action of the increased autoconversion and

accretion. This is illustrated by an additional simulation which uses only the enhancement for autoconversion, but ignores the effect on accretion. The resulting time series are much closer to the control run and show only a significant increase in rain rate at the very end of the simulation period. This underpins our results of the previous section that the rain formation in shallow cumulus clouds is not only limited by autoconversion, but also by accretion. Although accretion increases more strongly in

the Onishi kernel than in the Ayala-Wang kernel, the LES results show that this can not compensate for the weaker increase in autoconversion resulting in a reduced turbulence effect on rain formation. The main feedback of the different microphysical developments on the dynamics and the evolution of the boundary layer as a whole is that rain formation arrests the growth of the cloud layer as can be seen in the time series of the inversion height in Fig. 5, i.e., the Ayala-Wang kernel leads to a much shallower cloud layer in the precipitating regime. A similar behavior for different cloud droplet number densities was

shown by Stevens and Seifert (2008) and (Seifert et al., 2015). For the RICO case the boundary layer deepens and supports successively deeper clouds until moisture is efficiently removed by precipitation. Eventually the precipitating regime reaches a quasi-stationary state, the subsiding radiative-convective equilibrium (Seifert et al., 2015). This is also consistent with the finding that the enhancement of the warm rain process by taking into account turbulence effects on collisions has a very similar effect on cloud patterns, cloud fields, vertical profiles etc. as a change in the cloud droplet number.

The strong turbulence effect of both kernels suggested by Figure 5 is consistent with Seifert et al. (2010) and Wyszogrodzki et al. (2013), but two important aspects have to be considered. First, this behavior is transient, i.e., even the purely gravitational case would develop significant rain of order 1 mm/d after some time. Extending the simulation further shows that this happens after about 35 h. Second, Fig. 5 shows only simulations for a specific intermediate value of the cloud droplet number density. A lower value will make rain formation easier and more efficient also for the gravitational kernel and lead to smaller differences,

a higher droplet number may suppress precipitation even for the collection kernels that include turbulence effects. To get a more complete picture we have to discuss both effects.

### 5.3   Sensitivity to cloud droplet number

We have performed a larger set of large-eddy simulations for different cloud droplet number densities. In addition, simulations have been repeated with different random seeds to sample the stochastic uncertainty of the system and to reduce the standard

error in the statistical evaluation. Table 2 summarizes the results in terms of domain-mean statistical quantities like cloud cover, inversion height, rain water path, etc. As a measure for the temporal, i.e., transient behavior we have calculated two time scales that characterize the rain formation by the exceedance of thresholds for the domain-averaged rain rate, $t_1$ for a threshold of 0.1 mm/d and $t_2$ for 0.8 mm/d. While $t_1$ measures the first occurrence of rain at the surface, the larger threshold value of $t_2$ characterizes the transition to organized precipitation shallow convection (Seifert et al., 2015). The most important

results are summarized in Fig. 6 which illustrates the turbulence effects on the rain formation for different values of the cloud droplet number density. Shown are domain-mean quantities from 24 h to 30 h of the simulations and standard error is depicted by shaded areas. The standard error is estimated as $\sigma_x/n_x$ where $\sigma_x$ is the standard deviation of that variable and $n_x$ is it's effective sample size. For each simulation we estimate the effective sample size during the sampling period of 6 hours as $n_x = n_0(1-r_1)/(1+r_1)$ where $r_1$ is the lag-1 autocorrelation and $n_0$ is the number of samples in the time series. This simple

formulation gives almost the same results as a more sophisticated implementation following Zwiers and von Storch (1995). As shown in Fig. 6 rain water path and surface rain rate increase with decreasing cloud droplet number, but also show a pronounced impact of turbulence-induced collisions. For $N_c = 50\,\text{cm}^{-3}$, i.e., the simulations which are also shown in Fig. 5, both the Ayala-Wang kernel and the Onishi kernel lead to a strong increase in RWP and rain rate. For the lower value of $N_c = 35\,\text{cm}^{-3}$ the purely gravitational kernel used in the control simulations is sufficient to produce similar values of RWP and rain rate and the differences between the three kernels are no longer statistically significant. For an increase in droplet number the rain formation gets suppressed. Already for $N_c = 70\,\text{cm}^{-3}$ the rain rate and RWP for the Onishi kernel drops to values which are hardly different from the purely gravitational case, while the Ayala-Wang kernel still shows a strong enhancement leading to rain rates of order 1 mm/d during the 30 h period. Finally, for $N_c = 105\,\text{cm}^{-3}$ the rain formation starts to get suppressed even for the Ayala-Wang kernel and for droplet number exceeding that value all three collection kernels would only yield marginal precipitation within the 30 h period.

For low cloud droplet numbers we do not find a significant difference for the rain water path and the surface rain rate between the three different kernel during the 24 h to 30 h sampling period, because all three simulations develop a rain rate that is close to the quasi-equilibrium rain water flux. Nevertheless, the transient behavior is different between the three kernels for all droplet number densities as, e.g., seen from the time scales $t_1$ and $t_2$ in Figure 7. The Ayala-Wang kernel leads to an acceleration of the rain formation by more than 10 h for high drop number and still several hours for low droplet numbers. The acceleration caused by the Onishi kernel is less strong and becomes smaller for $t_2$ for low drop numbers while the difference in $t_1$ to the control run remains also for low drop numbers. This difference in the transient behavior leaves an imprint in the structure of the boundary layer even for long simulation times in the sense that the Ayala-Wang kernel, which develops rain most easily, arrests to growth of the boundary layer much earlier leading to the lowest inversion height in the precipitating regime (Fig. 6c). For the Onishi kernel this cloud macroscopic effect of the microphysical processes is much weaker. That the cloud droplet number and the microphysical efficiency of the cumulus clouds modulates the inversion height is consistent with the results of Stevens and Seifert (2008) and Seifert et al. (2015).

The turbulence effects on the collision rate, as postulated by the two different turbulence models, lead to a strong increase of the autoconversion rate and a moderate increase of accretion. This is true for both kernels, although the Onishi model has a weaker enhancement of autoconversion and a stronger increase in accretion, especially at high Reynolds numbers. It is therefore interesting to check whether a significant shift in the importance of those two warm rain processes can be observed in the large-eddy simulations. Figure 6d shows the ratio of accretion over autoconversion, $AC/AU$, for the sampling period of 24 h to 30 h. For all simulations accretion is the dominant process and total accretion exceeds autoconversion by a factor of 3 or more. Interestingly, the simulations which take into account turbulence effects show a higher $AC/AU$-ratio compared to the control simulations, which is counter-intuitive as the enhancement mostly affects autoconversion. This behavior can be understood from the relation between autoconversion and accretion. A higher autoconversion rate will most likely lead to a subsequent increase in accretion, because more small rain drops become available for accretional growth. Therefore an increase in the autoconversion rate, as caused by the turbulence effects, has little effect on the $AC/AU$-ratio. In fact, the higher

rain rate regimes of the simulations with the turbulence kernels favor accretion over autoconversion. Therefore the observed $AC/AU$-ratio is not directly linked to the turbulent enhancement factors of the process rates.

## 5.4 Sensitivity to grid resolution

Previous studies, e.g., by Matheou et al. (2011) and Seifert and Heus (2013) have emphasized that especially the precipitating RICO case exhibits a strong sensitivity to the grid spacing used in large-eddy simulations. We have therefore performed another set of simulations to test the sensitivity to grid spacing using 100 m, 50 m and 25 m horizontal mesh size for the three different collection kernels. The vertical grid spacing for all simulations is fixed at 25 m. Figure 8 summarizes the main results of the resolution study. The detailed statistics of the individual simulations are given in Table 3. For cloud liquid water path hardly any sensitivity to grid spacing is found, but the simulations with the Ayala-Wang kernel lead in general to a reduced CWP. This can be explained by the more rapid conversion of cloud water to rain, and by the shallower cloud layer in the precipitating regime. For rain water path and surface rain rate we find a strong increase with increasing resolution for the Onishi kernel and the control simulations. At 25 m grid spacing all three models give similar RWP and surface rain rate and differences are not statistically significant for those two variables. This is a similar behavior as for the reduced cloud droplet number. A small grid spacing in the LES makes the rain formation more rapid and the differences between the kernels becomes smaller when they all reach the precipitating regime before the chosen sampling period. This is confirmed by Fig. 9 which shows that the time scale $t_2$ decreases with resolution and at 25 m grid spacing all three kernels have a $t_2$ smaller than 20 h, i.e., the sampling period of 24 h to 30 h is in the precipitating regime for all three collision kernels. Figs. 8 and 9 reveal that the LES is not yet converged even at 25 m grid spacing. Unfortunately, higher resolution than the 25 m grid becomes very expensive and cannot be tested here. Differences in inversion height remain present even at the highest resolution, especially the Ayala-Wang kernel leads to much shallower cloud layers. A hint towards the causes of the strong resolution dependency is maybe given by the $AC/AU$-ratio which increases strongly for higher resolution. Especially the control run exhibits a significant increase from below 4 at 50 m grid spacing to almost 8 at 25 m. The rain efficiency, defined as the ratio of evaporation of rain over the sum of autoconversion and accretion, $1 - EV/(AU + AC)$, shows a behavior very similar to the $AC/AU$-ratio and suggests that the growth by accretion leads to large raindrops which are less susceptible to evaporation, thus more rain reaching the ground. The strong sensitivity of the rain formation to grid spacing may be surprising at first as individual precipitating cumulus clouds have horizontal scales of at least 1000 m and should be well resolved by the LES already at 50 m grid spacing. We suggest two possible mechanisms to explain the observed sensitivity. First, due to the strong nonlinearity of the autoconversion rate small scale fluctuations in cloud water may trigger autoconversion earlier and more often and initiate the rain formation more effectively at high resolution. Second, the in-cloud circulations which are better resolved at higher resolution increase the in-cloud residence time of the rain drops and therefore their overall growth by selfcollection and accretion. The latter effect has recently been emphasized as an important growth mechanism for raindrops in shallow cumulus clouds (Naumann and Seifert, 2016). Although it remains questionable whether a two-moment bulk scheme can represent recirculation properly, the strong increase of accretion observed in Fig. 8d would favor the second explanation. Whatever the detailed mechanism is, the strong sensitivity to grid spacing suggests that the larger modes of turbulence, like turbulent entraining eddies, which are resolved by

high-resolution LES, play an important role in enhancing the rain formation. This provides a second mechanism in addition to the effect of the small-scale isotropic turbulence on collision rates which is parameterized by the Ayala-Wang or Onishi kernel and sub-grid for any LES model.

## 6 Conclusions

We have derived a warm rain bulk two-moment scheme which incorporates the effects of small-scale isotropic turbulence on the collision rate following the two alternative models of Ayala-Wang and Onishi. The two collision kernels differ mostly in their Reynolds number dependency. While the Ayala-Wang model postulates an increase of autoconversion with Reynolds number, the Onishi model predicts a decrease of autoconversion, but an increase in accretion for high Reynolds number. The two newly derived variants of the Seifert-Beheng warm rain scheme have been tested and validated in 1D simulations and
compare favorably with the bin microphysics model that acts as a reference.

The new bulk scheme has been applied in large-eddy simulations of precipitating shallow convection to investigate the impact of the different collision kernels. Both turbulence kernels lead to a significant enhancement of the rain formation in shallow convective clouds, but the turbulence effect is much weaker for the Onishi kernel. Especially for intermediate cloud droplet numbers, in our simulations 50 cm$^{-3}$ but this might differ from case to case, the turbulence enhancement can lead to
15 a strong increase in rain water path and surface rain rate compared to a purely gravitational collection kernel. For the Ayala-Wang kernel we find a significant reduction of the height of the trade wind inversion, because the rapid rain formation arrests to growth of the cloud layer. This effect is not significant for the Onishi kernel. Overall, we found that the enhancement of the warm rain process by taking into account turbulence effects on collisions has a very similar effect on the evolution of the cloud field, the cloud patterns, and vertical profiles etc. as a corresponding change in the cloud droplet number.
The large-eddy simulations show a strong sensitivity to horizontal grid spacing with a more rapid rain formation at higher resolution. This suggests that the larger turbulent eddies like in-cloud circulations, which are resolved by high-resolution LES, can play an important role for the growth of rain drops. It is hypothesized that rain drops with large Stokes numbers, St$> 1$, can interact with these large turbulent eddies. For example, in the two-moment bulk scheme used in the present study such effects are not yet accurately parameterized and need to be investigated in more detail in future studies.
Our results show that the differences between the Ayala-Wang model and the Onishi models are significant and it needs to be clarified either by observations or by additional DNS studies which collision kernel is more realistic at high Reynolds numbers.

*Acknowledgements.* We thank the computing center of ECMWF where all simulations were performed using resources provided through DWD. We thank Ann Kristin Naumann for helpful comments on the manuscript. We thank the editor Graham Feingold and three anonymous reviewers for their comments that helped to improve the manuscript. The UCLA-LES model is distributed under GNU General Public License
and can easily be downloaded from https://github.com/uclales. Model code and input files necessary to reproduce the specific experiments of this study, are available from the corresponding author upon request.

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

**Table 1.** Coefficients as a result of the nonlinear regression for $k_{cc}$ as given by Eqs. (13)-(16).

| | Ayala-Wang | Onishi | Unit |
|---|---|---|---|
| $p$ | 1/4 | -1/8 | - |
| $a_1$ | $7.432 \times 10^{-4}$ | $3.985 \times 10^{-3}$ | $\mathrm{cm}^{-2}\mathrm{s}^3$ |
| $a_2$ | $-6.993 \times 10^{-5}$ | $6.210 \times 10^{-3}$ | $\mathrm{cm}^{-2}\mathrm{s}^3$ |
| $a_3$ | $-9.497 \times 10^{-2}$ | 1.331 | - |
| $b_1$ | 10.73 | 13.81 | $\mu\mathrm{m}$ |
| $b_2$ | 13.56 | 9.980 | $\mu\mathrm{m}$ |
| $b_3$ | 1.005 | 0.5018 | - |
| $c_1$ | 6.607 | 6.325 | $\mu\mathrm{m}$ |
| $c_2$ | 2.547 | -0.9238 | $\mu\mathrm{m}$ |
| $c_3$ | 0.2350 | -0.1528 | - |
| $\beta_{cc}$ | $3.480 \times 10^{-4}$ | $2.026 \times 10^{-3}$ | $\mathrm{cm}^{-2}\mathrm{s}^3$ |

**Table 2.** Statistics for the large-eddy simulations assuming different collection kernel. $N_x$ is the number of grid point in the horizontal, $\Delta x$ and $\Delta z$ are the horizontal and vertical grid spacing. Listed variables are the time scales $t_1$ and $t_2$ which characterize the transition to precipitating shallow convection (0.1 mm/d as rain rate-threshold for $t_1$, 0.8 mm/d for $t_2$, the area-averaged cloud cover $C$, the inversion height $z_i$, cloud liquid water path CWP in g/m$^2$, rain water path RWP in g/m$^2$, surface rain rate $R$ in $\mathrm{W\,m^{-2}}$ (29 $\mathrm{W\,m^{-2}}$ corresponds to mm d$^{-1}$). The ratio of accretion over autoconversion, $AC/AU$, and the rain efficiency, $RE = 1 - EV/(AU + AC)$ (both evaluated over the whole column). Time averages are from 24 h to 30 h. The simulations shown in Fig. 5 are indicated by a grey background. Simulations with identical model configuration (kernel, $N_x$, $\Delta x$, $\Delta z$, $N_c$) differ only by the random seed of the initial condition.

| $n$ | kernel | $N_x$ | $\Delta x$ | $\Delta z$ | $N_c$ | $t_1$ | $t_2$ | $C$ | $z_i$ | CWP | RWP | $R$ | AC/AU | RE |
|---|---|---|---|---|---|---|---|---|---|---|---|---|---|---|
| 1 | no turb. | 1024 | 50 | 25 | 35.0 | 7.5 | 21.6 | 16.7 | 2238 | 12.6 | 17.3 | 43.0 | 5.27 | 46.4 |
| 2 | no turb. | 1024 | 50 | 25 | 35.0 | 11.7 | 18.7 | 16.4 | 2245 | 12.8 | 15.9 | 37.8 | 4.98 | 42.7 |
| 3 | no turb. | 1024 | 50 | 25 | 50.0 | 19.6 | 31.9 | 16.0 | 2370 | 15.0 | 3.7 | 5.2 | 3.40 | 24.1 |
| 4 | no turb. | 1024 | 50 | 25 | 50.0 | 18.9 | 32.4 | 15.2 | 2375 | 14.9 | 4.1 | 5.9 | 3.42 | 23.2 |
| 5 | no turb. | 1024 | 50 | 25 | 50.0 | 21.3 | 34.9 | 15.6 | 2375 | 14.8 | 3.8 | 5.8 | 3.38 | 24.2 |
| 6 | no turb. | 1024 | 50 | 25 | 70.0 | 34.4 | 45.7 | 15.2 | 2388 | 15.4 | 1.1 | 1.4 | 2.98 | 18.7 |
| 7 | no turb. | 1024 | 50 | 25 | 70.0 | 28.5 | 43.7 | 15.4 | 2385 | 15.3 | 1.3 | 2.2 | 3.44 | 25.5 |
| 8 | no turb. | 1024 | 50 | 25 | 70.0 | 29.2 | 37.6 | 15.5 | 2385 | 15.6 | 1.4 | 2.1 | 3.49 | 23.3 |
| 9 | no turb. | 1024 | 50 | 25 | 105.0 | 46.0 | 50.5 | 15.2 | 2392 | 15.6 | 0.2 | 0.3 | 2.85 | 20.2 |
| 10 | Onishi | 1024 | 50 | 25 | 35.0 | 8.4 | 20.2 | 13.9 | 2213 | 10.7 | 18.0 | 41.8 | 5.11 | 43.5 |
| 11 | Onishi | 1024 | 50 | 25 | 35.0 | 6.2 | 17.7 | 14.4 | 2180 | 10.4 | 15.8 | 43.9 | 5.83 | 51.7 |
| 12 | Onishi | 1024 | 50 | 25 | 50.0 | 16.8 | 29.0 | 16.7 | 2351 | 15.0 | 9.4 | 17.0 | 4.45 | 32.4 |
| 13 | Onishi | 1024 | 50 | 25 | 50.0 | 13.0 | 25.7 | 18.1 | 2317 | 14.6 | 12.5 | 29.3 | 5.64 | 37.5 |
| 14 | Onishi | 1024 | 50 | 25 | 50.0 | 13.6 | 27.2 | 16.9 | 2337 | 15.4 | 13.4 | 30.1 | 5.34 | 40.8 |
| 15 | Onishi | 1024 | 50 | 25 | 50.0 | 12.8 | 25.1 | 19.4 | 2308 | 15.9 | 14.0 | 33.1 | 5.85 | 44.1 |
| 16 | Onishi | 1024 | 50 | 25 | 50.0 | 14.2 | 24.9 | 17.8 | 2295 | 15.0 | 16.3 | 42.5 | 6.31 | 46.8 |
| 16* | Onishi, au-only | 1024 | 50 | 25 | 50.0 | 16.8 | 28.8 | 16.3 | 2362 | 15.0 | 8.7 | 15.2 | 3.89 | 31.6 |
| 17 | Onishi | 1024 | 50 | 25 | 70.0 | 19.2 | 36.4 | 15.2 | 2370 | 14.8 | 3.0 | 4.6 | 4.00 | 24.9 |
| 18 | Onishi | 1024 | 50 | 25 | 70.0 | 21.7 | 38.2 | 15.3 | 2377 | 15.1 | 2.8 | 4.4 | 3.86 | 24.4 |
| 19 | Onishi | 1024 | 50 | 25 | 70.0 | 21.4 | 36.7 | 15.5 | 2377 | 15.2 | 2.9 | 4.4 | 3.88 | 25.4 |
| 20 | Onishi | 1024 | 50 | 25 | 70.0 | 24.0 | 33.5 | 15.8 | 2378 | 15.4 | 3.2 | 5.0 | 4.01 | 25.5 |
| 21 | Onishi | 1024 | 50 | 25 | 105.0 | 30.7 | 43.1 | 15.3 | 2392 | 15.5 | 0.9 | 1.6 | 4.37 | 27.7 |
| 22 | Ayala-Wang | 1024 | 50 | 25 | 35.0 | 4.8 | 13.6 | 10.5 | 2016 | 6.4 | 11.5 | 34.2 | 5.47 | 53.5 |
| 23 | Ayala-Wang | 1024 | 50 | 25 | 35.0 | 4.4 | 13.7 | 12.7 | 1901 | 7.6 | 14.1 | 46.2 | 6.68 | 62.6 |
| 24 | Ayala-Wang | 1024 | 50 | 25 | 50.0 | 5.6 | 17.8 | 13.6 | 2123 | 9.9 | 14.3 | 41.2 | 6.18 | 51.6 |
| 25 | Ayala-Wang | 1024 | 50 | 25 | 50.0 | 6.4 | 15.8 | 14.1 | 2091 | 9.7 | 15.2 | 48.3 | 7.82 | 61.3 |
| 26 | Ayala-Wang | 1024 | 50 | 25 | 50.0 | 6.1 | 18.0 | 15.0 | 2143 | 10.5 | 15.8 | 47.5 | 6.55 | 55.6 |
| 27 | Ayala-Wang | 1024 | 50 | 25 | 50.0 | 7.2 | 18.2 | 14.0 | 2151 | 10.4 | 15.3 | 41.8 | 5.82 | 48.3 |
| 28 | Ayala-Wang | 1024 | 50 | 25 | 70.0 | 13.7 | 26.2 | 16.4 | 2309 | 14.0 | 12.7 | 30.2 | 5.54 | 41.8 |
| 29 | Ayala-Wang | 1024 | 50 | 25 | 70.0 | 9.7 | 22.0 | 17.8 | 2265 | 13.5 | 15.3 | 42.7 | 6.63 | 49.6 |
| 30 | Ayala-Wang | 1024 | 50 | 25 | 70.0 | 10.6 | 21.4 | 17.5 | 2244 | 13.2 | 14.6 | 42.0 | 6.65 | 50.3 |
| 31 | Ayala-Wang | 1024 | 50 | 25 | 105.0 | 19.3 | 35.2 | 15.9 | 2364 | 15.1 | 4.7 | 9.5 | 4.95 | 33.9 |

**Table 3.** As previous Table, but for the simulations to investigate the resolution dependency at $N_c = 50 \text{ cm}^{-3}$.

| $n$ | kernel | $N_x$ | $\Delta x$ | $\Delta z$ | $N_c$ | $t_1$ | $t_2$ | $C$ | $z_i$ | CWP | RWP | $R$ | AC/AU | RE |
|---|---|---|---|---|---|---|---|---|---|---|---|---|---|---|
| 1 | no turb. | 2048 | 25 | 25 | 50.0 | 7.4 | 15.4 | 13.2 | 2072 | 10.4 | 10.7 | 33.1 | 8.00 | 56.4 |
| 2 | no turb. | 2048 | 25 | 25 | 50.0 | 7.9 | 20.4 | 17.1 | 2195 | 14.0 | 13.5 | 38.2 | 6.72 | 51.0 |
| 3 | no turb. | 2048 | 25 | 25 | 50.0 | 7.7 | 16.3 | 15.0 | 2052 | 10.8 | 14.3 | 47.0 | 9.01 | 61.6 |
| 4 | no turb. | 1024 | 50 | 25 | 50.0 | 19.6 | 31.9 | 16.0 | 2370 | 15.0 | 3.7 | 5.2 | 3.40 | 24.1 |
| 5 | no turb. | 1024 | 50 | 25 | 50.0 | 18.9 | 32.4 | 15.2 | 2375 | 14.9 | 4.1 | 5.9 | 3.42 | 23.2 |
| 6 | no turb. | 1024 | 50 | 25 | 50.0 | 21.3 | 34.9 | 15.6 | 2375 | 14.8 | 3.8 | 5.8 | 3.38 | 24.2 |
| 7 | no turb. | 512 | 100 | 25 | 50.0 | 24.1 | 46.7 | 12.4 | 2422 | 12.8 | 2.8 | 3.4 | 2.88 | 16.1 |
| 8 | Onishi | 2048 | 25 | 25 | 50.0 | 7.3 | 17.3 | 14.4 | 2066 | 11.1 | 13.8 | 44.8 | 8.05 | 59.5 |
| 9 | Onishi | 2048 | 25 | 25 | 50.0 | 6.2 | 16.0 | 14.6 | 2062 | 10.3 | 12.9 | 42.0 | 8.92 | 61.4 |
| 10 | Onishi | 1024 | 50 | 25 | 50.0 | 16.8 | 29.0 | 16.7 | 2351 | 15.0 | 9.4 | 17.0 | 4.45 | 32.4 |
| 11 | Onishi | 1024 | 50 | 25 | 50.0 | 13.0 | 25.7 | 18.1 | 2317 | 14.6 | 12.5 | 29.3 | 5.64 | 37.5 |
| 12 | Onishi | 1024 | 50 | 25 | 50.0 | 13.6 | 27.2 | 16.9 | 2337 | 15.4 | 13.4 | 30.1 | 5.34 | 40.8 |
| 13 | Onishi | 1024 | 50 | 25 | 50.0 | 12.8 | 25.1 | 19.4 | 2308 | 15.9 | 14.0 | 33.1 | 5.85 | 44.1 |
| 14 | Onishi | 1024 | 50 | 25 | 50.0 | 14.2 | 24.9 | 17.8 | 2295 | 15.0 | 16.3 | 42.5 | 6.31 | 46.8 |
| 15 | Onishi | 512 | 100 | 25 | 50.0 | 16.0 | 33.7 | 13.0 | 2398 | 12.5 | 7.4 | 12.3 | 3.96 | 26.8 |
| 16 | Ayala-Wang | 2048 | 25 | 25 | 50.0 | 4.7 | 12.7 | 10.6 | 1939 | 7.3 | 10.2 | 34.7 | 7.79 | 59.1 |
| 17 | Ayala-Wang | 1024 | 50 | 25 | 50.0 | 5.6 | 17.8 | 13.6 | 2123 | 9.9 | 14.3 | 41.2 | 6.18 | 51.6 |
| 18 | Ayala-Wang | 1024 | 50 | 25 | 50.0 | 6.4 | 15.8 | 14.1 | 2091 | 9.7 | 15.2 | 48.3 | 7.82 | 61.3 |
| 19 | Ayala-Wang | 1024 | 50 | 25 | 50.0 | 6.1 | 18.0 | 15.0 | 2143 | 10.5 | 15.8 | 47.5 | 6.55 | 55.6 |
| 20 | Ayala-Wang | 1024 | 50 | 25 | 50.0 | 7.2 | 18.2 | 14.0 | 2151 | 10.4 | 15.3 | 41.8 | 5.82 | 48.3 |
| 21 | Ayala-Wang | 512 | 100 | 25 | 50.0 | 6.1 | 22.9 | 11.4 | 2321 | 9.5 | 14.5 | 31.2 | 4.48 | 36.8 |

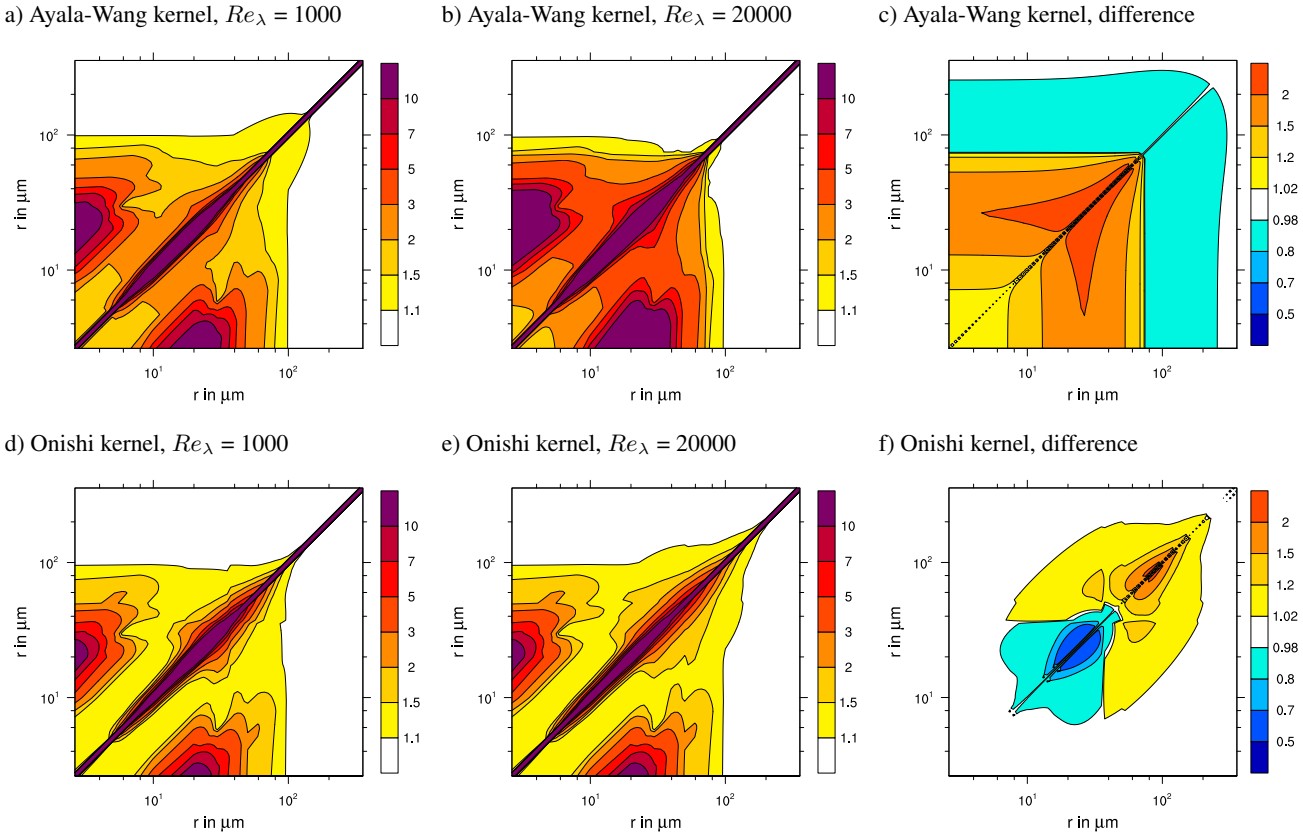

**Figure 1.** Enhancement factor of the collision-coalescence kernel for a dissipation rate of $\epsilon = 1000$ cm$^2$s$^{-3}$. Shown are (a) the Ayala-Wang kernel for a Taylor-microscale Reynolds number of 1000, (b) the Ayala-Wang kernel for $Re_\lambda = 20000$, (c) the ratio of the Ayala-Wang kernel at $Re_\lambda = 20000$ and $Re_\lambda = 1000$. The second row show the same plot for the Onishi kernel at $\epsilon = 1000$ cm$^2$s$^{-3}$ and (d) $Re_\lambda = 1000$, (e) $Re_\lambda = 20000$ and (f) the ratio between the kernels at those two Reynolds numbers.

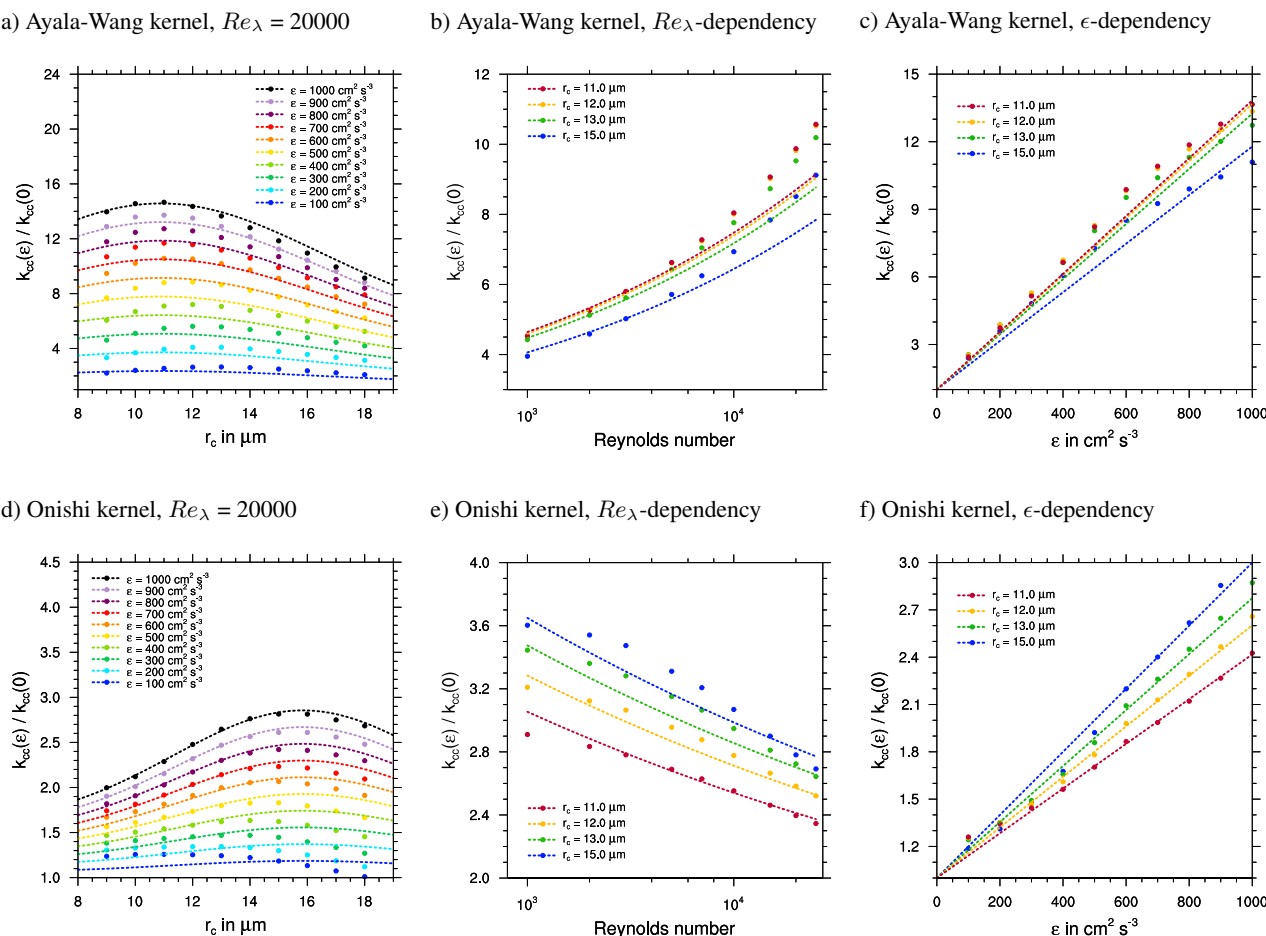

**Figure 2.** Enhancement factor of the autoconversion rate for the Ayala-Wang kernel (upper row) and the Onishi kernel (lower row) at at $Re_\lambda = 20000$ (a,c), the Reynolds number dependency of the enhancement factor at $\epsilon = 600\ \mathrm{cm^2 s^{-3}}$ (b,d), and the dependency on dissipation rate for $Re_\lambda = 20000$ (c,f). Data points (dots) are based on numerical solutions of the stochastic collection equation (SCE), the parameterization shown (dashed lines) is Eq. (10) with the coefficients as given in Table 1. All plots are shown for $\nu = 1$. Note the different scaling of the y-axis for both kernels.

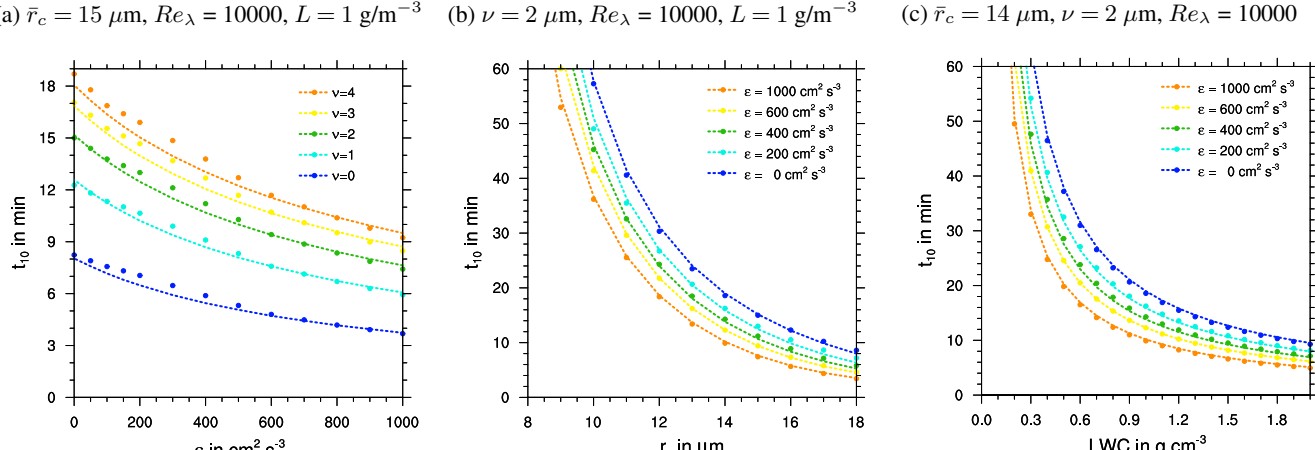

**Figure 3.** Time $t_{10}$, that is needed to convert 10 % of the initial cloud water to rain water (a) $t_{10}$ as a function of dissipation rate $\epsilon$ for various $\nu$ (and $\bar{r}_c = 15$ $\mu$m, $Re_\lambda = 10000$), (b) $t_{10}$ as a function of mean cloud droplet radius $\bar{r}_c$ for various values of dissipation rate $\epsilon$ (and $\nu = 2$, $Re_\lambda = 10000$) and (c) $t_{10}$ as a function of the initial cloud liquid water content for various values of dissipation rate $\epsilon$ (and $\bar{r}_c = 14$, $\nu = 2$, $Re_\lambda = 10000$). Data points are numerical solution of the SCE, dashed lines represent the solutions of the two-moment bulk scheme with the enhancement factor for autoconversion based on the Onishi kernel as given by Eq. (10) and the coefficients of Table 1.

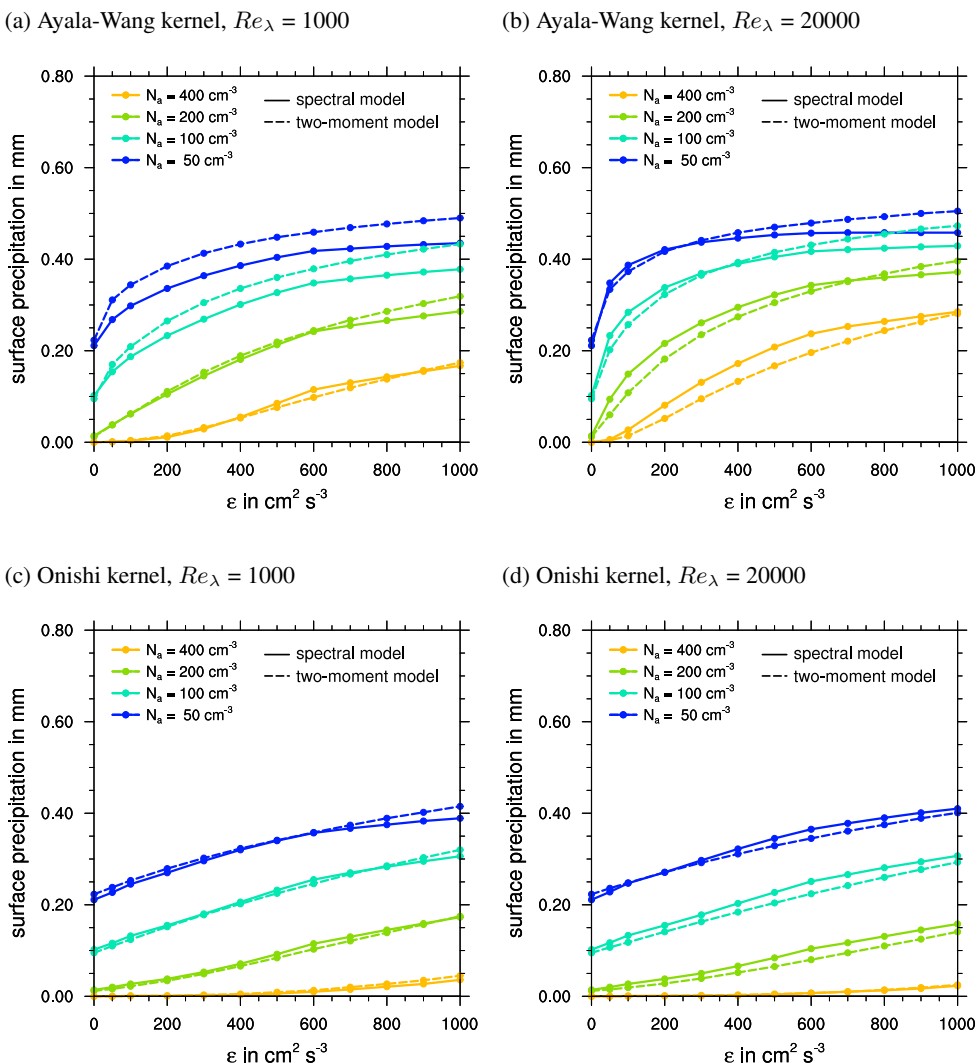

**Figure 4.** Accumulated surface precipitation of the 1D kinematic model as a function of the assumed in-cloud turbulent dissipation rate $\epsilon$ (other parameters are temperature gradient $\Gamma_0 = 1.5$ K/km, the maximum updraft speed $w_0 = 2$ m/s, and the updraft time scale $\tau_w = 40$ min). Shown are results from the Ayala-Wang model at $Re_\lambda = 1000$ (a) and $Re_\lambda = 20000$ (b), as well as the Onishi model at those two Reynolds numbers (c,d). Results of the spectral bin reference model are depicted with solid lines, the results of the two-moment parameterization with dashed lines.

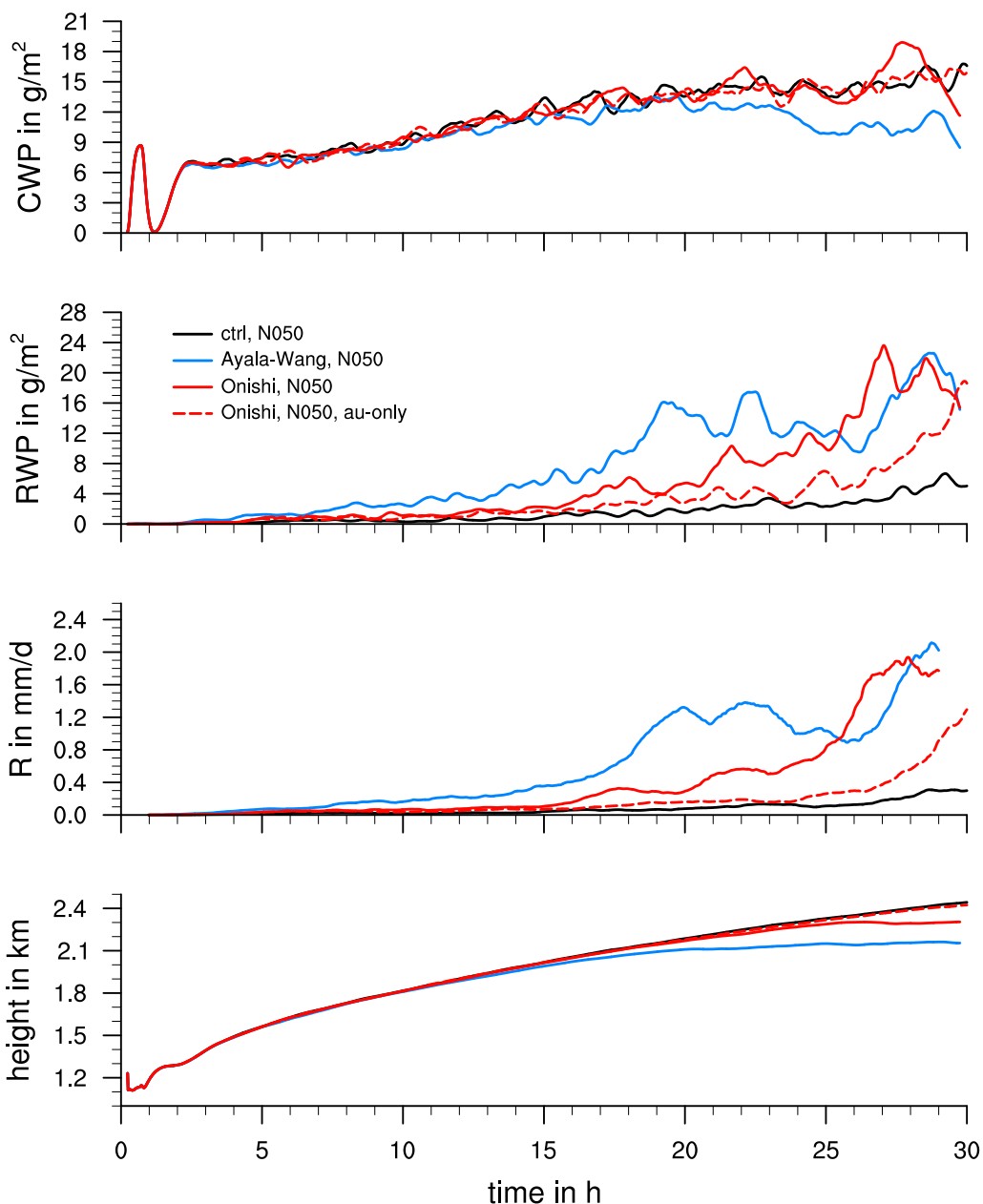

**Figure 5.** Time series of the cloud liquid water path, rain water path, the surface rain rate and the inversion height for four simulations using the three different collection kernels. The simulation marked 'au-only' applies the turbulent enhancement only to autoconversion, but ignores the effect on accretion. We have applied a running average to all time series with an averaging window of 120 min for the surface rain rate and 30 min for RWP, CWP and inversion height.

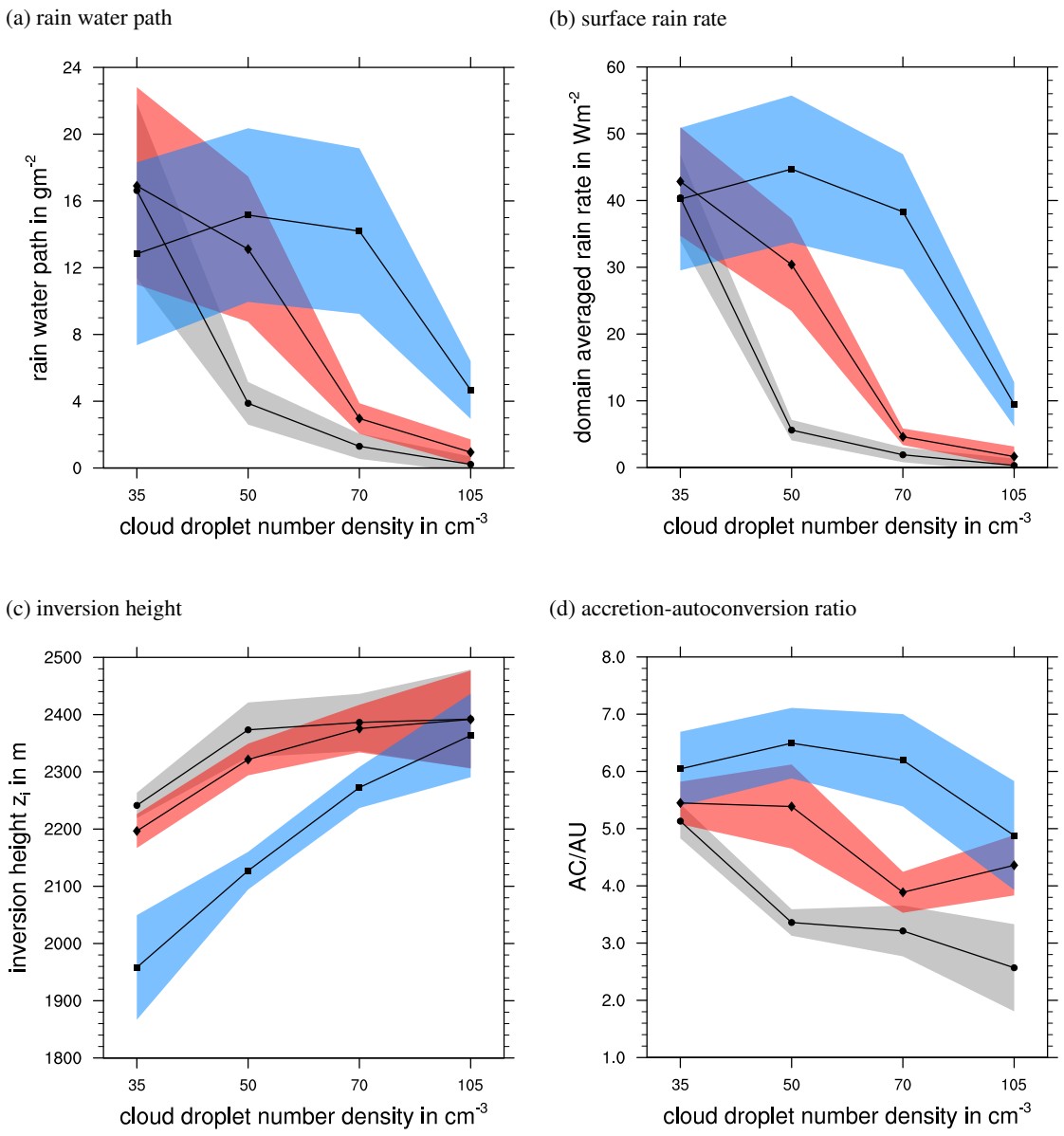

**Figure 6.** Sensitivity of LES results to variations in the cloud droplet number density. Shown are the rain water path, surface rain rate, inversion height, and the accretion-autoconversion ratio for the three different collection kernels of the control simulations using the purely gravitational kernel (bullets, grey shading), the Ayala-Wang kernel (squares, blue shading), and the Onishi kernel (diamonds, red shading). The shaded area indicates the standard error at a 95 % confidence level.

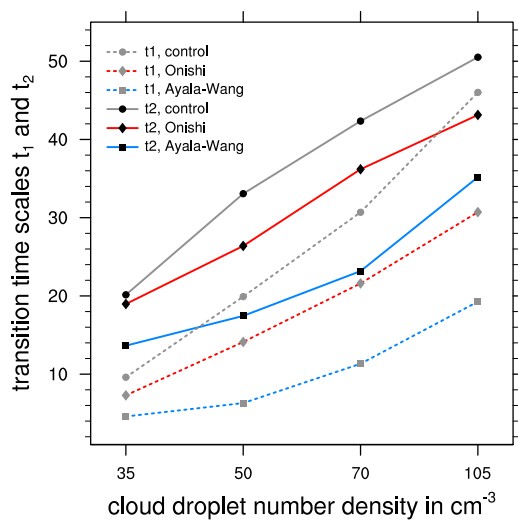

**Figure 7.** Transition time scales $t_1$ (dashed, grey symbols) and $t_2$ (solid, black symbols) defined as the time when the domain-averaged rain rate exceeds 0.1 mm/d or 0.8 mm/d, respectively, for the first time. The transition times are averaged over multiple simulations with different random seeds.

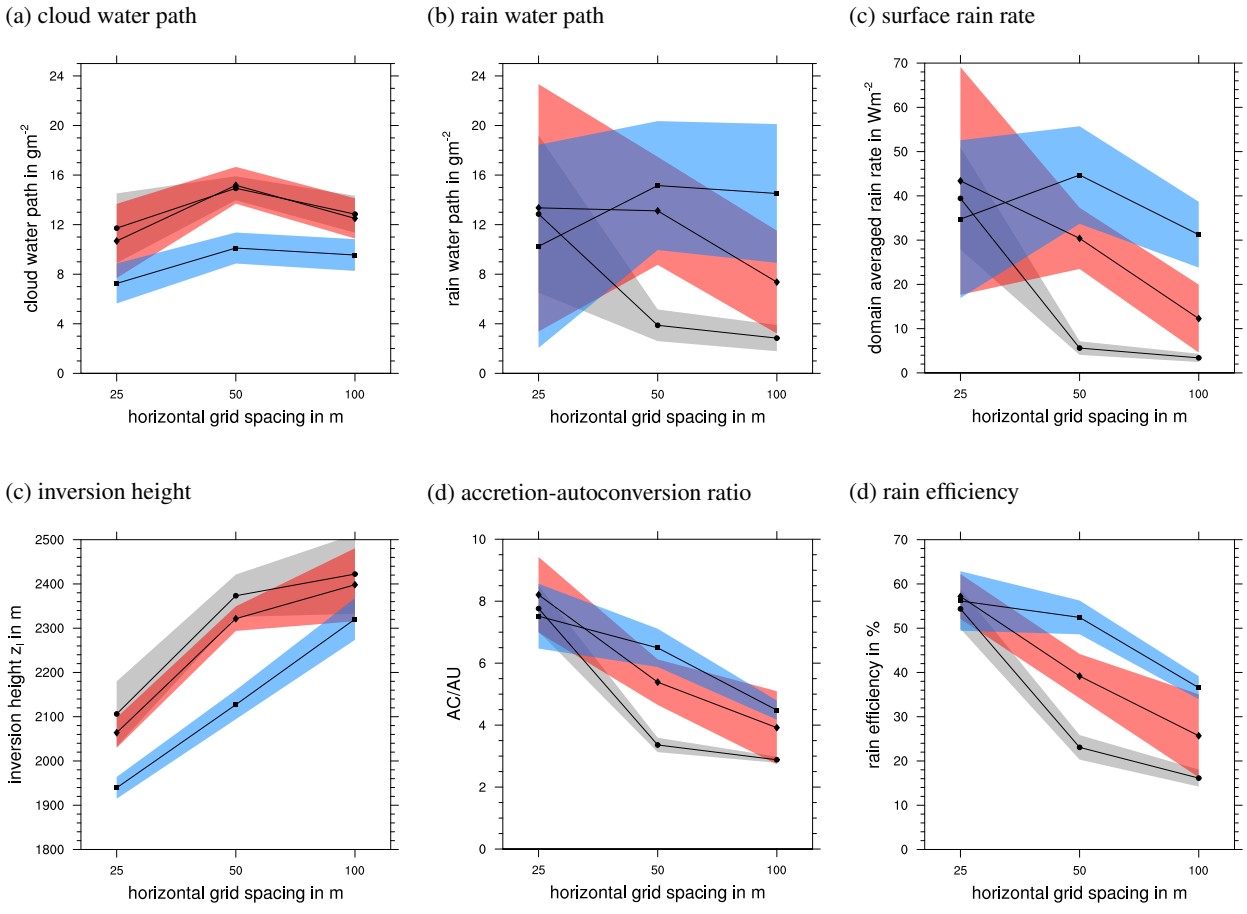

**Figure 8.** As Fig. 6, but showing the dependency of the results in the sampling period 24 h to 30 h on grid spacing for a cloud droplet number density of $N_c = 50 \, \text{cm}^{-3}$.

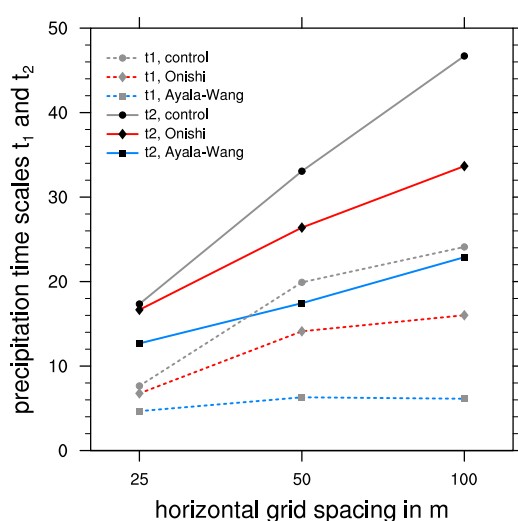

**Figure 9.** As Figure 7, but showing the dependency of the rain formation time scales $t_1$ and $t_2$ on horizontal grid spacing for a cloud droplet number density of $50\,\mathrm{cm}^{-3}$.