# Peer review of "Turbulence effects on warm rain formation in precipitating shallow convection revisited"

_Atmospheric Chemistry and Physics, 2016_

## Referee Comment (RC1) · Anonymous Referee #1 · 22 Jun 2016

Review of "Turbulence effects on warm rain formation in precipitating shallow convection revisited" by Seifert and Onishi

Recommendation: minor revision

This manuscript, in essence a follow-up to Seifert et al. (QJ, 2010), compares results of LES simulations of shallow precipitating convection focusing on the effects of two collision kernels that include effects of small-scale turbulence, the Ayala-Wang kernel and the Onishi kernel. This is a well-written paper than can be published as is, but I have several specific comments that the authors might consider to further improve their presentation.

[Figure]

Specific comments (the first two more serious):

1. The logic (the same as applied in Seifert et al. 2010) is to derive the autoconversion and accretion enhancements for the 2-moment scheme of Seifert and Beheng, and then to use the modified 2-moment scheme in LES simulations. I feel this is a justifiable methodology (especially considering the expense of the bin scheme), but I feel the 1D kinematic model of Seifert and Stevens might not be sufficient to validate the 2-moment implementation. To me, the key difference between bin and 2-moment scheme is the representation of droplet sedimentation (mass/number weighted in the 2-moment scheme and different for every bin in the bin scheme). Thus, the surface rainfall (e.g., Fig. 4 in the current manuscript) may agree well in the 1D test, but may differ more significantly in a test where horizontal variability is included, for instance, in a 2D kinematic test. Overall, I feel the difference in the sedimentation between bulk and bin schemes deserves a closer look, not necessary in the context of the current paper, but in a more general study. I would like to see this aspect at least to be recognized in the current draft.

2. The fact that differences in the cloud microphysics (i.e., rain formation in the current study) may affect cloud dynamics is obvious. However, this aspect is not even mentioned in the current manuscript except for (relatively obscure and not discussed) references to the inversion height shown in Fig. 6. I think some discussion of the feedback from the microphysics to the cloud field dynamics (e.g., deepening of the cloud field that is an unfortunate feature of the RICO setup) should be added to the manuscript. Overall, separation of purely microphysical effects from the impact on cloud dynamics is difficult, but needs to be done to fully understand the impacts. Again, I feel just mentioning this issue and leaving it for a future study (perhaps applying the "piggybacking" method that Grabowski used in his studies published in JAS in 2014 and 2015) would be sufficient. A hint of the dynamic feedback can perhaps be shown by adding the inversion height to time evolutions shown in Fig. 5.

3. P. 3, paragraph starting at l. 30. The way enhancements are shown in Fig. 1 does

not allow seeing the enhancement for droplets of equal (or very close) size. Can you show the enhancement for equal-size droplets for the two formulations? How important are such collisions for the acceleration of rain formation?

4. P. 4, the end of section 4. I think you can explicitly say when discussing Fig. 4 that the differences are about 10-20% max, a relatively small difference considering differences seen in cloud field simulations.

5. P. 7, discussion around l. 29. I think the discussion has to do with the undesirable aspect of the RICO case, namely, the deepening of the cloud field. Perhaps this should be openly stated (I think it is not obvious to someone not familiar with the RICO case). My suggestion at the end of 2 above would also help to make this obvious.

6. P. 8, text between l. 10 and 15. I feel more explanation is needed here. What is sigma_x (mean standard deviation from the time average?). What is the "lag-1 auto-correlation"? How many samples are there in the 6-hour time series? This method of assessing statistical significance is different from the Student t-test statistic, correct?

7. P. 10, l. 30. Here is an example of the microphysics-dynamics feedback that is important in this problem, yet it is really not discussed in the current draft.

---

## Referee Comment (RC2) · Anonymous Referee #2 · 24 Jul 2016

The authors propose a new parameterization of warm rain formation, using collection kernels which account for turbulence intensity and including turbulence properties in autoconversion parameterization. Then they compare the new parameterization using 1-D bin model of warm microphysics and finally apply to LES of cumulus convection, using two different collection kernels documented in the literature. The main results show a remarkable dependency of the simulations result on the collection kernel applied. In the conclusions the authors underline necessity of more observations and DNS studies.

The results, which suggest that our knowledge of collection coalescence in warm rain process is not sufficient to unambiguously implement into LES are not surprising, yet

valuable. The paper is written clearly, quality of the presentation is excellent. However, in the opinion of the reviewer, there are several points which should be discussed or analyzed in more detail, especially in the context of the overall negative conclusion of the study.

In particular:

Why enhancement factors for autoconversion and time t10 are presented for Onishi kernel only? How they differ for Ayala-Wang kernel? Accumulated surface precipitations in 1D for both kernels agree with the proposed parameterization, but are very different. This additional analysis, supplementing that of Onishi and Seifert (2016) discussed in the present text would be of value.

Analysis of LES results is insufficient. In particular, the authors discuss basic microphysical and cloud field parameters between 24 and 30 hours of simulations (Figs. 6 and 8) without paying sufficient attention to cloud patterns, cloud fields, vertical profiles. In effect information on the effects of proposed parameterization / collection kernels on convection dynamics is partially missing. Figure 5 suggests that for several cases there is a significant variability within the last hours of the simulations, which is confirmed in transition times presented in Fig. 7. Extended discussion of the differences would add to the paper.

---

## Referee Comment (RC3) · Anonymous Referee #3 · 2 Aug 2016

**Turbulence effects on warm rain formation in precipitating shallow convection revisited, by Axel Seifert and Ryo Onishi**

In this paper, the authors followed the previous work of Seifeit et al. (2010) to study the effect of turbulence on the evolution of warm rain in precipitating shallow convection. Their main purpose is to compare two alternative formulations of turbulent collision kernel, Ayala-Wang and Onishi. The reference case of purely gravitational collection is also considered. They first developed an approximate autoconversion parameterization based on the turbulent collision kernels. The main conclusion is that the results of precipitation rate, characteristic rain development times, etc., depend on the kernel, calling for further investigation of turbulent kernel formulation. This conclusion is reached based on a large set of LES runs which considered different initial droplet number density, shape (mean size), and LES grid resolution.

The paper is interesting and may be published. However, some clarifications need to be made in order to provide a more complete and fair picture in the context of the complex problem of rain initiation.

1. The starting point of the paper is the introduction of two collection kernels. Fig. 1 shows how the Reynolds number affects the change of kernel in each case. For most regions, the Ayala-Wang kernel seems to have a more stronger Reynolds number effect. Is there any region in Fig. 1 f) showing a stronger dependence compared to Fig. 1 c). If so, can the reason be provided?

2. The paper relies heavily on the contents in other papers including basic definitions. For example, the precise definitions of autoconversion, accretion, and selfcollection are not given. It would be useful to provide definitions of such.

3. Furthermore, regarding the enhancement of accretion and selfcollision $k_{rr}$, I assume this factor is used in determining the mean size of rain drops. Can an equation like Eq. (8) be provided to show how $k_{rr}$ is actually incorporated in the moment methods.

4. One of the observations is that the Ayala-Wang kernel lead to faster autoconversion and Onishi's leads to faster accretion. The faster autoconversion is due to stronger $Re$ dependence. Can the reason for faster accretion for the Onishi's kernel be provided? This could be discussed in terms of aspects related to the point 1 above.

5. The study uses a single mass ($2.6 \times 10^{-10} kg$ or about 40 $\mu m$ in radius) as the dividing size between cloud droplets and rain drops. I wonder how this choice affects the conclusions of the paper. Can the authors study other dividing size such as 25 $\mu m$ or 35 $\mu m$ as the dividing size? This is important since a very rough moment method is used in the LES.

6. The formulation involves a shape parameter $\nu$ (Eq. 7). I assume $A$ and $B$ are related to $L_c$ and $\bar{x}_c$. It is not clear if $\nu$ is kept as a constant during the LES simulation and how $\nu$ is determined. Can this be clarified?

7. In the model equation (10), a single exponent $p$ is used for the whole range of $Re$. In reality, the collection kernel (specifically the RDF) first increases with $Re$, then saturates or decreases slowly with $Re$. The question is then how valid a single exponent in representing the effect of flow $Re$.

8. Another observation is that the Ayala-Wang kernel leads to shallow inversion height. However, in Wyszogrodzki et al. (2013) and Grabowski et al. (2015, Atmos. Chem. Phys., 15: 913-926) based on the spectral bin method, it is shown the dynamic effect of faster droplet growth is a deep cloud top. I wonder if these two are contradictory, and if the reason for this contradiction is due to their use of the moment method. Clearly, the strong sensitivity of the collision kernel with droplet size and shape of droplet size distribution requires a more accurate representation than the two-moment method. The authors should clarify the various errors associated with the moment method, and potential effect on the conclusions of the paper.

---

## Author Comment (AC1) · 1 Sep 2016

**REPLY TO REVIEWER #1:**

We thank the reviewer for the comments that helped us to improve the manuscript.

**1. The logic (the same as applied in Seifert et al. 2010) is to derive the autoconversion and accretion enhancements for the 2-moment scheme of Seifert and Beheng, and then to use the modified 2-moment scheme in LES simulations. I feel this is a justifiable methodology (especially considering the expense of the bin scheme), but I feel the 1D kinematic model of Seifert and Stevens might not be sufficient to validate the 2-moment implementation. To me, the key difference between bin and 2-moment scheme is the representation of droplet sedimentation (mass/number weighted in the 2-moment scheme and different for every bin in the bin scheme). Thus, the surface rainfall (e.g., Fig. 4 in the current manuscript) may agree well in the 1D test, but may differ more significantly in a test where horizontal variability is included, for instance, in a 2D kinematic test. Overall, I feel the difference in the sedimentation between bulk and bin schemes deserves a closer look, not necessary in the context of the current paper, but in a more general study. I would like to see this aspect at least to be recognized in the current draft.**

We agree with the reviewer that a 2D framework would be a much better test and fully agree with the statement concerning sedimentation. We have added a sentence at the beginning of section 4 reading

*Although the 1D model provides a reasonable idealized framework for such a test, we would recommend to use a kinematic 2D model (e.g. Szumowski et al. 1998, Morrison and Grabowski 2007) in future studies, because the 1d framework might not be sensitive enough to differences in the treatment of sedimentation which are more relevant in a more complex flow field. Here we apply the simpler 1D model for consistency with Seifert et al. (2010).*

**2. The fact that differences in the cloud microphysics (i.e., rain formation in the current study) may affect cloud dynamics is obvious. However, this aspect is not even mentioned in the current**

manuscript except for (relatively obscure and not discussed) references to the inversion height shown in Fig. 6. I think some discussion of the feedback from the microphysics to the cloud field dynamics (e.g., deepening of the cloud field that is an unfortunate feature of the RICO setup) should be added to the manuscript. Overall, separation of purely microphysical effects from the impact on cloud dynamics is difficult, but needs to be done to fully understand the impacts. Again, I feel just men- tioning this issue and leaving it for a future study (perhaps applying the piggybacking method that Grabowski used in his studies published in JAS in 2014 and 2015) would be sufficient. A hint of the dynamic feedback can perhaps be shown by adding the inversion height to time evolutions shown in Fig. 5.**

The deepening of the cloud layer is one of the most interesting features of the RICO case and makes it especially valuable when investigating the effects of cloud microphysics on the evolution of the cloud layer. The effect of different microphysical choices or assumptions on the boundary layer dynamics has been extensively discussed by Stevens and Seifert (2008), van Zanten et al. (2011), Seifert et al. (2015) and others. Therefore we have not discussed this in detail in the current manuscript. In the revised version we follow the recommendation of the reviewer and have added the inversion height to Fig. 5 and included a few sentences in section 5.2. reading

*The main feedback of the different microphysical developments on the dynamics and evolution of the boundary layer as a whole is that rain formation arrests the growth of the cloud layer as it can be seen in the time series of the inversion height in Fig. 5, i.e., the Ayala-Wang kernel leads to a much shallow cloud layer in the precipitating regime. A similar behavior for different cloud droplet number densities was shown by Stevens and Seifert (2008). For the RICO case the boundary layer deepens and supports successively deeper clouds until moisture is efficiently removed by precipitation. Eventually the precipitating regime reaches a quasi-stationary state, the subsiding radiative-convective equilibrium (Seifert et al., 2015).*

Using the piggybacking methodology would be an attractive alternative to our extensive LES study. Without piggybacking the randomness of the individual LES runs makes it actually necessary to use ensembles of LES realisations, which is computationally very demanding. We agree with the reviewer that piggybacking offers an attractive method to overcome such problems. Nevertheless, we refrained from using the method because it leads to inconsistencies between the dynamics and the microphysics and the results have to be interpreted very carefully. The old fashioned brute force approach used in our study is maybe less elegant, but each simulation is physically fully consistent. Nevertheless, we fully agree that such studies as presented in our manuscript could benefit from the piggybacking approach, if it is carefully used and interpreted.

**3. P. 3, paragraph starting at l. 30. The way enhancements are shown in Fig. 1 does not allow seeing the enhancement for droplets of equal (or very close) size. Can you show the enhancement for equal-size droplets for the two formulations? How important are such collisions for the acceleration of rain formation?**

The enhancement factor for equal-size droplets is by definition infinite. We would refer to Fig. 4 and section 4.3 of the accompanying paper by Onishi and Seifert (2016, ACP) for a discussion of the collision frequency of similar sized droplets. We think that such collisions, e.g. selfcollection events of small raindrops, are very important especially in maritime clouds with low to moderate cloud droplet numbers and relatively high autoconversion rate. In such clouds small drizzle drops can be present in abundance, but their growth is relatively slow due to the low to moderate cloud water content (limiting accretion) and the rare collisions between similar sized drops (limiting selfcollection). As soon as some drops grow due to some selfcollection events, they also have an advantage in accretion due to the larger fall speed of a bigger drop. Such a chain of processes is what we postulate to explain the increase in accretion rate (Eq. 15), which is stronger than the enhancement of the kernel itself for the accretion process. Or in other words: The enhancement of the collision rate of similar-sized drops leads to a modification of the drop size distribution (a stronger tail) due to selfcollection which is part of the enhancement of accretion parameterized by Eq. (15).
The importance of selfcollection for the surface rain rate in maritime shallow cumulus is also discussed in the recent paper by Naumann et al. (2016) by applying a detailed diagnostics using a Lagrangian drop model (aka superdroplets).

**4. P. 4, the end of section 4. I think you can explicitly say when discussing Fig. 4 that the differences are about 10-20 % max, a relatively small difference considering differences seen in cloud field simulations.**

Figure 4 is not only there to show that the bulk scheme works reasonably well, but also and maybe more important to discuss the differences between the two collection kernels. It is not clear to which of the two the reviewer refers. The difference between the Ayala-Wang kernel and the Onishi kernel can actually be a factor of 2 (for moderate dissipation rates).

**5. P. 7, discussion around l. 29. I think the discussion has to do with the undesirable aspect of the RICO case, namely, the deepening of the cloud field. Perhaps this should be openly stated (I think it is not obvious to someone not familiar with the RICO case). My suggestion at the end of 2 above would also help to make this obvious.**

Following the recommendation of the reviewer, we have included a discussion of the deepening of the cloud layer in section 5.2. Nevertheless, we do not understand why the deepening of the cloud field should be 'undesirable'. As long as the subsidence drying is not able to compensate the moisture input from the latent heat flux the cloud layer has to grow. We could agree with the statement that the growth of the cloud layer is artifically slow in the RICO case making it much more susceptible to microphysical perturbations than a boundary layer in which local radiative cooling leads to a more rapid equilibration of the cloud layer, i.e., the deepening should be much more efficient than in the standard RICO case used here.

**6. P. 8, text between l. 10 and 15. I feel more explanation is needed here. What is $\sigma_x$ (mean standard deviation from the time average?). What is the lag-1 auto- correlation? How many samples are there in the 6-hour time series? This method of assessing statistical significance is different from the Student t-test statistic, correct?**

Yes, the domain mean quantities are simple time series and $\sigma_x$ is the standard deviation as it is explained in the text. The standard deviation of a

time series is always 'the mean standard deviation from the time average'. The lag-1 autocorrelation of a discrete time series is the autocorrelation between subsequent samples of that time series. This is standard terminology in statistics and time series analysis (and easily found in most textbooks). Software packages like R, Matlab, NCL, etc. provide functions to calculate these quantities. The estimation of the effective sample size is a classic problem in statistics and the reference provided in the paper gives a more detailed discussion of this topic.

The number of independent samples depends on the quantity, because different variables have different autocorrelation time scales. For the rain rate the effective sample size in a 6-hour time series is between 3 and 10 with an average of about 6. This makes sense as a shallow convective rain event has a typical duration (or time scale) of 1 hour. For the inversion height the sample size is only 1 per 6-hour time series, because the inversion height is the result of the combined action of all boundary layer eddies (i.e. all clouds), i.e., each LES run provides only 1 independent estimate for the inversion height. Due to this averaging property the standard deviation of the inversion height is also much smaller and consequently the standard error is small although the effective sample size is only 1 per LES run. Knowing the effective sample size is a prerequisite for the Student t-test, but we decided to plot only the standard error and not to delve deeper into test for statistical significance. We would argue that even without doing statistical hypothesis testing our analysis is still more elaborate than what is usually presented when comparing different LES runs.

**7. P. 10, l. 30. Here is an example of the microphysics-dynamics feedback that is important in this problem, yet it is really not discussed in the current draft.**

This feedback is now mentioned several times in the revised manuscript. For a detailed discussion of the basic behavior we refer to the literature, e.g., Stevens and Seifert (2008) as well as Seifert et al. (2015).

---

## Author Comment (AC2) · 1 Sep 2016

**REPLY TO REVIEWER #2:**

We thank the reviewer for the comments that helped us to improve the manuscript.

**Why enhancement factors for autoconversion and time t10 are presented for Onishi kernel only? How they differ for Ayala-Wang kernel? Accumulated surface precipita- tions in 1D for both kernels agree with the proposed parameterization, but are very different. This additional analysis, supplementing that of Onishi and Seifert (2016) dis- cussed in the present text would be of value.**

In the revised version we have included the corresponding plots for the Ayala-Wang kernel and extended the discussion of the enhancement factor for the autoconversion rate.

*The different autoconversion enhancement factors for the two kernels and the quality of the fits is shown by Fig. 2 in which also the Reynolds number dependency is shown in more detail. The results for the Ayala-Wang kernel show somewhat higher enhancement factors compared to Seifert et al. (2010), mostly due to the improved treatment of the collision efficiency (cf. Onishi and Seifert 2016). The Onishi kernel shows much lower enhancement factors and the maximum is shifted to larger (mean) droplet radii compared to the Ayala-Wang kernel. The $Re_\lambda$-dependency reveals that especially for the Onishi kernel the value of the exponent, $p = -1/8$, is really just a fit with limited physical meaning as the actual slope has significant dependencies on $\bar{r}_c$ and $Re_\lambda$. This more complicated behavior is consistent with the analysis presented by Onishi and Seifert (2016) who showed that the Reynolds number dependency of the kernel varies with Stokes number (e.g. their Figure 2). For the Ayala-Wang kernel the numerical data shows a steeper increase with $Re_\lambda$ compared to the parameterization. This is mostly because we kept the exponent at $p = 1/4$ as in Seifert et al. (2010), although the extended range of the dissipation rate in the current study would ask for a slightly higher exponent. The dependency on dissipation rate is assumed to be linear in Eq. (10) and this is confirmed for the Onishi kernel, but for the Ayala-Wang kernel the $\epsilon$-dependency becomes weaker for high dissipation rates.*

**Analysis of LES results is insufficient. In particular, the authors**

**discuss basic micro- physical and cloud field parameters between 24 and 30 hours of simulations (Figs. 6 and 8) without paying sufficient attention to cloud patterns, cloud fields, vertical profiles. In effect information on the effects of proposed parameterization / collection kernels on convection dynamics is partially missing. Figure 5 suggests that for several cases there is a significant variability within the last hours of the simulations, which is confirmed in transition times presented in Fig. 7. Extended discussion of the differences would add to the paper.**

The different assumptions for the collection kernel and the resulting modification of the warm rain process do not fundamentally change the behavior of the cloud dynamics, i.e., an enhancement of the warm rain process by taking into account turbulence effects on collisions has a very similar effect on cloud patterns, cloud fields, vertical profiles etc. as a change in the cloud droplet number. The latter experiments have been extensively described and discussed in the literature, e.g., by Stevens and Seifert (2008), van Zanten et al. (2011), Seifert and Heus (2013), Seifert et al. (2015) and others. Therefore we present only those aspects of the simulations which help us to learn something new and gain a deeper understanding of the interaction of turbulence and warm rain processes. An example is the response of the accretion-autoconversion ratio to the different kernel assumptions discussed in section 5.3 and 5.4. Specific aspects of the cloud dynamics for the turbulence effects, like the fact that the highest dissipation rates are observed near cloud top, are already discussed in Seifert et al. (2010) and Wyszogrodzki et al. (2013) and there is no reason to repeat this in the current manuscript. A more detailed analysis of the resolved in-cloud turbulence and its effect on rain formation would be very interesting and, in our opinion, new, but is beyond the scope of the current manuscript.

---

## Author Comment (AC3) · 1 Sep 2016

**REPLY TO REVIEWER #3:**

We thank the reviewer for the comments that helped us to improve the manuscript.

**1. The starting point of the paper is the introduction of two collection kernels. Fig. 1 shows how the Reynolds number affects the change of kernel in each case. For most regions, the Ayala-Wang kernel seems to have a more stronger Reynolds number effect. Is there any region in Fig. 1 f) showing a stronger dependence compared to Fig. 1 c). If so, can the reason be provided?**

For the details of the collection kernels we refer to the accompanying paper by Onishi and Seifert (2016, ACP). The Onishi kernel shows a strong Reynolds number dependency for small regions in Fig. 1f, mostly along the diagonal, i.e., for droplets of similar size. This is due to the Reynolds number dependency of the optimal Stokes number for the preferential concentration effect. It is postulated that the optimum value for the preferential concentration shifts from $St = 1$ to slightly higher Stokes numbers for high Taylor-microscale Reynolds numbers. Due to the fact that the flanks of the radial distribution function $g_{11}$, which quantifies the preferential concentration effect, are quite steep a shift in $g_{11}$ leads to a strong increase (or decrease) in a narrow range of drop sizes (Stokes numbers). This is basically what we see in Figure 1f.

**2. The paper relies heavily on the contents in other papers including basic definitions. For example, the precise definitions of auto-conversion, accretion, and selfcollection are not given. It would be useful to provide definitions of such.**

Yes, this paper is intended for scientists who are familiar with the basic concepts of cloud physics, bulk microphysical parameterizations and turbulence effects on collision rates. It is hardly possible to review all those topics in a scientific paper. Nevertheless, we provide a short introduction of essential definitions and relations for particle-laden turbulence in section 2. For the basic ideas and definition of warm rain bulk microphysics we would like to refer to Klaus Beheng's review paper

*Beheng, K. D.: The evolution of raindrop spectra: A review of basic microphysical essentials, Rainfall: State of the Science, Geophys. Monogr., 191, 2948, 2010.*

but following the request of the reviewer we have extended the introduction of the bulk microphysics scheme at the beginning of section 3, which now provides a short introduction to bulk microphysics parameterizations.

**3. Furthermore, regarding the enhancement of accretion and self-collision krr, I assume this factor is used in determining the mean size of rain drops. Can an equation like Eq. (8) be provided to show how krr is actually in- corporated in the moment methods.**

In the revised version of the manuscript these equations are explicitly given in section 4. This actually helped to fix some minor inconsistencies in the presentation.

**4. One of the observations is that the Ayala-Wang kernel lead to faster autoconversion and Onishis leads to faster accretion. The faster autoconver- sion is due to stronger Re dependence. Can the reason for faster accretion for the Onishis kernel be provided? This could be discussed in terms of aspects related to the point 1 above.**

This is discussed in detail in the accompanying paper by Onishi and Seifert (2016, ACP). As already explained in the answer to question 1, the main effect is the shift of the preferential concentration optimum to higher Stokes numbers in case of high Taylor-microscale Reynolds numbers. The larger Stokes numbers correspond to larger drops which belong to the raindrop category of the bulk scheme.

**5. The study uses a single mass ($2.6 \times 10^{-10}$ kg or about 40 $\mu$m in radius) as the dividing size between cloud droplets and rain drops. I wonder how this choice affects the conclusions of the paper. Can the authors study other di- viding size such as 25 $\mu$m or 35 $\mu$m as the dividing size? This is important since a very rough moment method is used in the LES**.

The threshold size is not arbitrarily chosen, but corresponds to the minimum of the bi-modal mass distribution function during the evolution of the drop size distribution (see e.g. Fig. 4 of Beheng's review paper). A small change like using 35 $\mu$m instead of 40 $\mu$m will not affect the results as this will only change the autoconversion rate by about 10 %. A reduction to 25 $\mu$m is inconsistent with the assumptions made in SB2001 and simply too small for the separating size of a two-category scheme. To explicitly predict the formation of such small drizzle drops the three-category scheme of Sant et al. (2013, J. Atmos. Sci) could be used instead. For shallow cumulus clouds this is not necessary, but it might be interesting for stratocumulus.

**6. The formulation involves a shape parameter (Eq. 7). I assume A and B are related to Lc and xc . It is not clear if is kept as a constant during the LES simulation and how is determined. Can this be clarified?**

Yes, the gamma shape parameter of the cloud droplet distribution $\nu$ is constant during an LES simulation. We mention this explicitly in the revised manuscript in section 5.1. The meaning of this constant $\nu$ in the SB2001 scheme is often misunderstood as it is actually only the shape parameter before coagulation kicks in. It would be possible to estimate a local time-dependent $\nu$ which is consistent with the assumption of the SB2001 model from the universal function $\Phi_{au}(\tau)$ or simply as a function of $\tau$. Here $\tau$ is the non-dimensional internal time variable of the system, which describes the evolution of the cloud droplet distribution due to coagulation. The autoconversion rate, Eq. (8), is not simply the solution of the collision integral for a fixed $\nu$, but includes the change in the drop size distribution and, hence, an evolving $\nu$. On the other hand, the cloud droplet distribution is not strictly a gamma distribution during coagulation, and therefore estimating $\nu$ would provide only limited information (especially the tail of the distribution is much more important than the shape described by $\nu$).

**7. In the model equation (10), a single exponent p is used for the whole range of Re. In reality, the collection kernel (specifically the RDF) first increases with Re, then saturates or decreases slowly with Re. The question is then how valid a single exponent in representing the effect of flow Re.**

Given the various approximations and uncertainties in the kernel, the bulk

scheme and the LES model, and the sensitivity to grid resolution of the LES, we would argue that the use of a single exponent $p$ to describe the Re-dependency is a minor problem. For a true reference simulation we would need an LES model that can predict $\epsilon$ and $\mathrm{Re}_\lambda$ independently. Having such a model we could then think about using a super-droplet approach to simulate the coagulation explicitly with the full Onishi kernel.

**8. Another observation is that the Ayala-Wang kernel leads to shallow inversion height. However, in Wyszogrodzki et al. (2013) and Grabowski et al. (2015, Atmos. Chem. Phys., 15: 913-926) based on the spectral bin method, it is shown the dynamic effect of faster droplet growth is a deep cloud top. I wonder if these two are contradictory, and if the reason for this contradiction is due to their use of the moment method. Clearly, the strong sensitivity of the collision kernel with droplet size and shape of droplet size distribution requires a more accurate representation than the two-moment method. The authors should clarify the various errors associated with the moment method, and potential effect on the conclusions of the paper.**

This is maybe related to the simulation of the BOMEX case (by Wyszogrodzki et al. 2013 and Grabowski et al. 2015) vs the RICO case that is used here and has been used in Seifert et al. (2010). In addition the domain used by Wyszogrodzki et al. (2013) is quite small with only 6.4 km in the horizontal compared to 51.2 km in the current study, and the simulated time period is only 6 h in Wyszogrodzki et al. (2013) compared to at least 30 h in the current study. For example, the small domain may be dominated by individual clouds which can lead to a different interpretation of the results. Maybe more important, the BOMEX case was initially set up as a non-precipitating case and the system is in equilibrium without the formation of precipitation (Siebesma et al. 2003, JAS). Hence, the formation rain leads to a perturbation of this quasi-equilibirium state and pushes the system into an instationary transient state. In contrast, RICO is not in equilibrium without rain as it was designed based on data from a rainy period. The RICO case approaches a quasi-equilibrium only due to the formation of rain late in the simulations. These differences in the model setup and the case design can lead to quite different behavior and different interpretations. It would be very interesting to do an intercomparison with both cases using spectral bin and bulk methods. Unfortunately, it is very expensive and time consuming to do large sensitivity studies with bin microphysics models. Comparing just a few simulations is very questionable due to the strong sensitivity and randomness of precipitating shallow convection, especially in the RICO case.

We have tried our best to convince the reader that the moments method provides a reasonable parameterization of the collision-coalescence process, e.g., with help of Figs. 3 and 4. A full quantification of the errors is beyond the scope of the paper. Nevertheless, we are confident that the results are qualitiatively meaningful and provide valuable insights in the behavior of precipitating shallow convection and the turbulence effect on rain formation. As suggested by the the analysis presented in section 5.4 the largest uncertainty of the model might actually be associated with the still too coarse resolution of the LES model.